# Structural mechanisms of TRPM7 activation and inhibition

Kirill D. Nadezhdin [1], Leonor Correia[2], Chamali Narangoda [3], Dhilon S. Patel[3], Arthur Neuberger [1], Thomas Gudermann [2,4], Maria G. Kurnikova [3] ✉, Vladimir Chubanov [2] ✉ & Alexander I. Sobolevsky [1] ✉

The transient receptor potential channel TRPM7 is a master regulator of the organismal balance of divalent cations that plays an essential role in embryonic development, immune responses, cell mobility, proliferation, and differentiation. TRPM7 is implicated in neuronal and cardiovascular disorders, tumor progression and has emerged as a new drug target. Here we use cryo-EM, functional analysis, and molecular dynamics simulations to uncover two distinct structural mechanisms of TRPM7 activation by a gain-of-function mutation and by the agonist naltriben, which show different conformational dynamics and domain involvement. We identify a binding site for highly potent and selective inhibitors and show that they act by stabilizing the TRPM7 closed state. The discovered structural mechanisms provide foundations for understanding the molecular basis of TRPM7 channelopathies and drug development.

The melastatin-type transient receptor potential (TRPM) ion channels are involved in numerous biological processes, including oxidative stress, cell death, mineral homeostasis, and regulation of vascular tone, and represent targets for treatment of neurodegenerative disorders, cardiovascular diseases, diabetes, cancer, and pain[1–3]. Among eight TRPM channels (TRPM1-8), the first group (TRPM2/4/5/8) comprises either nonselective cation-permeable ion channels (TRPM2/8) or monovalent cations-selective channels (TRPM4/5)[1,2]. Intracellular $Ca^{2+}$ represents a common positive regulator of TRPM2/4/5/8 channels and recent cryo-EM analysis provided mechanistic insights into this regulation and gating[4–9]. The second group (TRPM1/3/6/7) includes channels that are highly permeable to divalent cations, with the structure of TRPM3 being recently solved in a GPCR-inhibited conformation[10]. TRPM7 is the most studied member of this group, which in many regards epitomizes biophysical and physiological characteristics of TRPM1/3/6 as well[11–14]. Each TRPM7 subunit has a C-terminal protein kinase domain, which phosphorylates several protein substrates, including nuclear histones[15–22]. The channel of TRPM7 displays constitutive activity, which is tightly regulated by several endogenous factors, including $Mg^{2+}$, Mg-ATP and $PIP_2$[23–28].

TRPM7 represents a master regulator of the cellular balance of divalent cations and mediates uptake of $Zn^{2+}$, $Mg^{2+}$ and $Ca^{2+}$, which is required for the normal prenatal development and healthy adulthood[29–34]. Point mutations in TRPM7 are linked to hypomagnesemia with secondary hypocalcemia, trigeminal neuralgia, $Mg^{2+}$-dependent macrothrombocytopenia, and Guamanian neurodegenerative disease[35–39]. Pharmacological regulation of TRPM7 can benefit patients with immune and cardiovascular disorders, tissue fibrosis and tumours[3,11–13] and several small-molecule TRPM7 inhibitors and activators have been already examined preclinically[40,41].

TRPM7 was previously characterized using cryo-electron microscopy (cryo-EM)[42]. Despite the three available 3.3–4.1 Å-resolution structures of the protein purified in detergent micelles were determined in different ionic conditions, all three captured the channel in the same closed conformation, thus, providing limited insight into the regulatory mechanisms of TRPM7[42]. Here, we embark on structural characterization of TRPM7 purified in both detergent and

[1]Department of Biochemistry and Molecular Biophysics, Columbia University, New York, NY, USA. [2]Walther-Straub Institute of Pharmacology and Toxicology, LMU Munich, Munich, Germany. [3]Chemistry Department, Carnegie Mellon University, Pittsburgh, PA, USA. [4]Comprehensive Pneumology Center, German Center for Lung Research (DZL), Munich, Germany. ✉e-mail: kurnikova@cmu.edu; vladimir.chubanov@lrz.uni-muenchen.de; as4005@cumc.columbia.edu

lipid nanodiscs and solve 2.17–2.99 Å-resolution cryo-EM structures in the closed apo state, open conducting conformations activated by a gain-of-function mutation or by an agonist and inhibited non-conducting states bound to highly potent TRPM7 antagonists.

## Results

### Structures of TRPM7 in the closed apo state

To produce protein for our structural studies, we used a C-terminally-truncated mouse TRPM7 construct, which represents the functional channel with the proteolytically cleaved kinase domain[42]. First, we determined structures of TRPM7 in the apo state by subjecting the protein purified in glyco-diosgenin (GDN) detergent or reconstituted in lipid nanodiscs to cryo-EM. The corresponding 3D reconstructions resulted in 2.61-Å and 2.19-Å resolution structures (Supplementary Table 1 and Supplementary Figs. 1–3). Despite the two structures being nearly indistinguishable, the cryo-EM map for the protein reconstituted in lipid nanodiscs (Fig. 1a, b) had better overall quality and resolution than the one for the protein purified in GDN (Supplementary Table 1 and Supplementary Fig. 2). Therefore, we mainly use the structure of TRPM7 in lipid nanodiscs in our further analysis.

The TRPM7 structure represents a tetramer assembled of four identical subunits (Fig. 1c), with the overall architecture reminiscent of other TRPM channels[4,5,7,43–46]. Each TRPM7 subunit consists of an intracellular N-terminal domain (NTD, residues 1–789), transmembrane domain (TMD, 839–1146), and cytosolic C-terminal domain (CTD, 1157–1230) (Supplementary Fig. 4a, b). The NTD includes four melastatin homology regions (MHR1-4). The TMD resembles TMDs in voltage-gated and TRP channels and includes six transmembrane helices (S1–S6) and a pore loop (P-loop) between S5 and S6. The first

four helices form a bundle called S1–S4 domain or voltage sensing-like domain. S5, P-loop, and S6 comprise the pore domain that connects to the S1–S4 domain in a subunit-swapped manner. The TMD ends with the amphiphilic TRP helix, which runs almost parallel to the membrane surface and represents a signature of the TRP channel family. The CTD follows the TRP helix and includes rib and pole helices. The pole helices of four TRPM7 subunits assemble a coiled-coil in the middle of the intracellular domain.

Due to the high quality and resolution of the cryo-EM reconstructions, our apo-state structures provide more complete models of TRPM7, including the regions that were not resolved in the previously published structures (Supplementary Fig. 4c, d)[42]. Since all previously published TRPM7 structures represented a single closed state, the nearly perfect superposition of their resolved regions with the corresponding regions in our structures (Supplementary Fig. 4d) suggests that our apo-state structures also represent the same closed state, TRPM7_Closed.

Our apo-state structures have higher quality and resolution compared to the previously published closed-state TRPM7 structures[42], which provide an additional benefit. We observed excellent protein density in the TMD (Supplementary Fig. 4e) surrounded by additional density, which allowed us to build models for 18 lipids per TRPM7_Closed subunit (Fig. 1a, c). Most of these densities have a head-and-two-tails appearance and most likely represent phospholipids. However, two types of putative lipid densities had a shape drastically different from phospholipids, and we modeled them as cholesterol and GDN (Fig. 1d and Supplementary Fig. 4f). Since we did not introduce cholesterol during protein purification, the corresponding densities in our structure most probably represent endogenous cholesterol molecules that

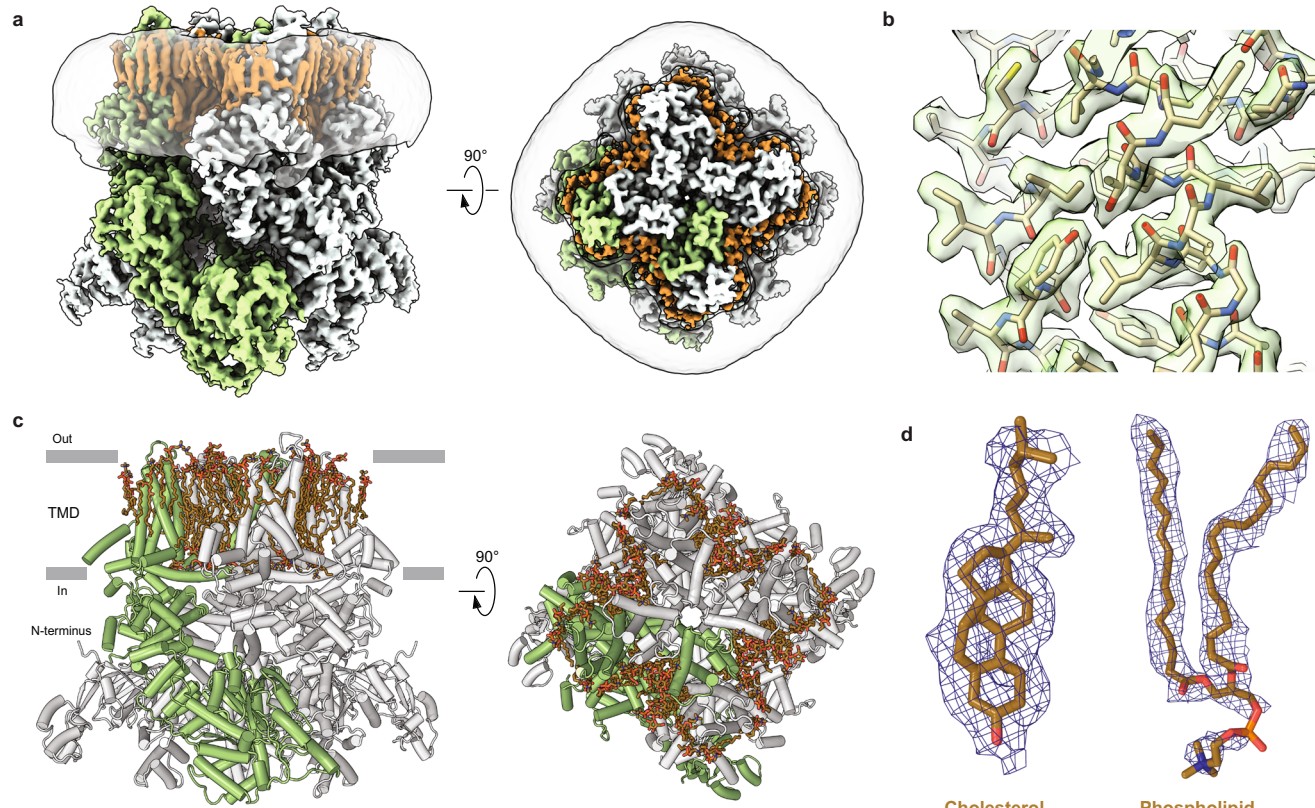

**Fig. 1 | Cryo-EM structure of TRPM7 in the apo state. a** Cryo-EM density map of TRPM7 in the closed, apo state at 2.19 Å resolution, viewed parallel to the membrane (left) and extracellularly (right). One of the four subunits is colored green, and the other three in grey. Lipid densities are highlighted in brown. The semi-transparent surface represents the lipid nanodisc density. **b** Close-up view of a map region in the TMD. **c** Structural model of TRPM7_Closed reconstituted in lipid nano-discs. One of the four subunits is colored in green, and the other three in grey. Lipids are shown as brown sticks. **d** Representative cryo-EM density (blue mesh) and models (brown sticks) for cholesterol and phospholipid.

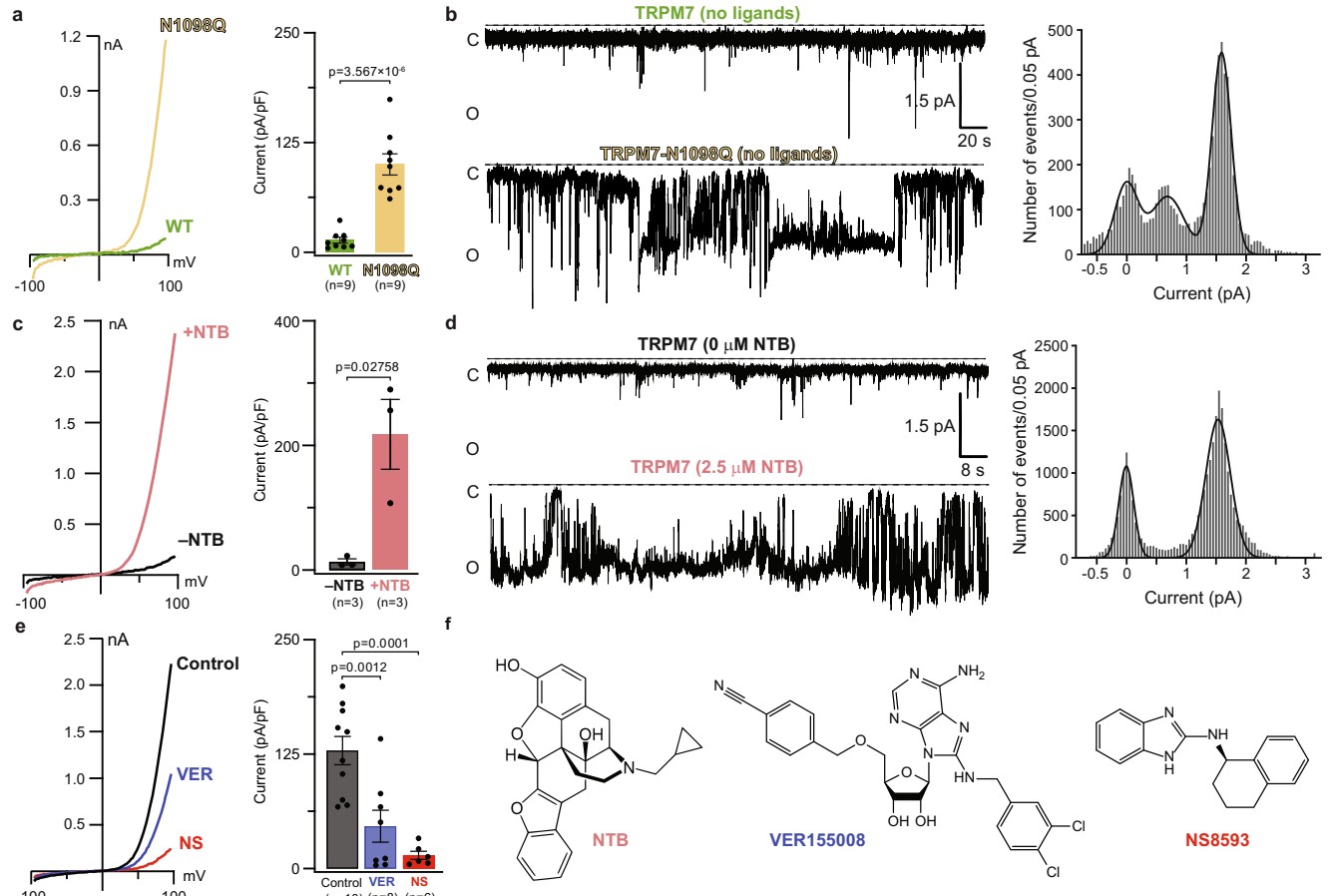

**Fig. 2 | Function of wild-type TRPM7 and the gain-of-function mutant N1098Q.**
**a**, **c**, **e** Whole-cell patch-clamp recordings made from HEK 293T cells expressing wild-type (WT) (**a**, **c**, **e**) or mutant N1098Q (**a**) TRPM7 in response to a −100 to +100 mV voltage ramp. Left panels show representative current–voltage (I–V) relationships immediately after establishing the whole-cell configuration (**a**), without addition or after 170-s application of 500 µM NTB (**c**) and after 450-s exposure to the standard external solution in the absence (Control) or presence of 10 µM VER or 10 µM NS, following saturation of currents after 150 s of recordings (**e**). Right panels show current amplitudes measured at +80 mV (mean ± SEM). *n* is

the number of cells. Statistical comparison was made using unpaired *t* test (**a**, **c**) or ordinary one-way ANOVA (**e**). Significance was accepted at $P \leq 0.05$.
**b**, **d** Representative single-channel currents recorded at the membrane potential of −100 mV from WT (**b**, **d**) or N1098Q mutant (**b**) TRPM7 reconstituted in lipid bilayers in the absence (**b**, **d**) or presence (**d**) of 2.5 µM NTB. The corresponding all-points amplitude histograms from three independent traces for TRPM7-N1098Q mutant (**b**) and for TRPM7 in the presence of 2.5 µM NTB (**d**) are shown on the right. The black curves are fits with three (**b**) or two (**d**) Gaussians. **f** Chemical structures of naltriben (NTB), VER155008 and NS8593.

interact with the channel in physiological conditions. On the other hand, GDN was used during the extraction step of purification and likely remains attached to the protein despite the following GDN-free reconstitution into lipid nanodiscs.

## Opening of TRPM7 with the gain-of-function mutation N1098Q

To study TRPM7 gating, we introduced the gain-of-function mutation N1098Q[47]. The high constitutive activity of this version of TRPM7 can be readily detected in patch-clamp recordings before the depletion of the negative regulator of the channel, intracellular free magnesium[47]. Indeed, immediately after establishing the whole-cell configuration, currents recorded from HEK 293T cells expressing TRPM7-N1098Q had much higher amplitude than cells transfected with wild-type TRPM7 (Fig. 2a). To compare the activity of TRPM7-N1098Q and TRPM7 at the single-channel level, we purified both proteins and reconstituted them into planar lipid bilayers. In the absence of activating ligands, TRPM7 showed infrequent openings, with the maximum open probability, $P_o = 0.02 \pm 0.01$ ($n = 3$; 1207 events), indicating a low level of the channel's spontaneous activity (Fig. 2b). In contrast, the gain-of-function TRPM7-N1098Q mutant showed increased single-channel activity, characterized by the higher value of $P_o = 0.63 \pm 0.30$ ($n = 3$; 11,449 events). At the same time, the single-channel current

amplitude for TRPM7-N1098Q remained approximately the same as for TRPM7 (Fig. 2b), indicating that the mutation increased the frequency of channel opening rather than the single-channel conductance. Our experiments, therefore, confirmed that TRPM7-N1098Q forms a functional ion channel, which remains constitutively open for a substantial fraction of time.

To capture the open conformation of TRPM7, we purified the gain-of-function TRPM7-N1098Q mutant protein and subjected it to cryo-EM analysis. The three-dimensional cryo-EM reconstruction of TRPM7-N1098Q yielded a 2.46-Å resolution structure, with the architecture grossly similar to TRPM7$_{Closed}$ (Fig. 3a). Nonetheless, these structures were substantially different, as evidenced by the high value of the root-mean-square deviation (r.m.s.d. = 1.53 Å for Cα atoms) for their overall superposition. While the S1–S4 and coiled-coil domains in this superposition appear to overlap perfectly, the clear difference between the two structures originates in the pore domain, emphasized by shifts in the positions of TRP helices, and is further propagated to the N-terminal domains, which move as rigid bodies away from the axis of the overall 4-fold rotational symmetry.

Since the most significant conformational changes were observed in S6 and the TRP helix, representing the key elements of the TRP channel gating machinery[4,5,48,49], we compared the ion-conducting

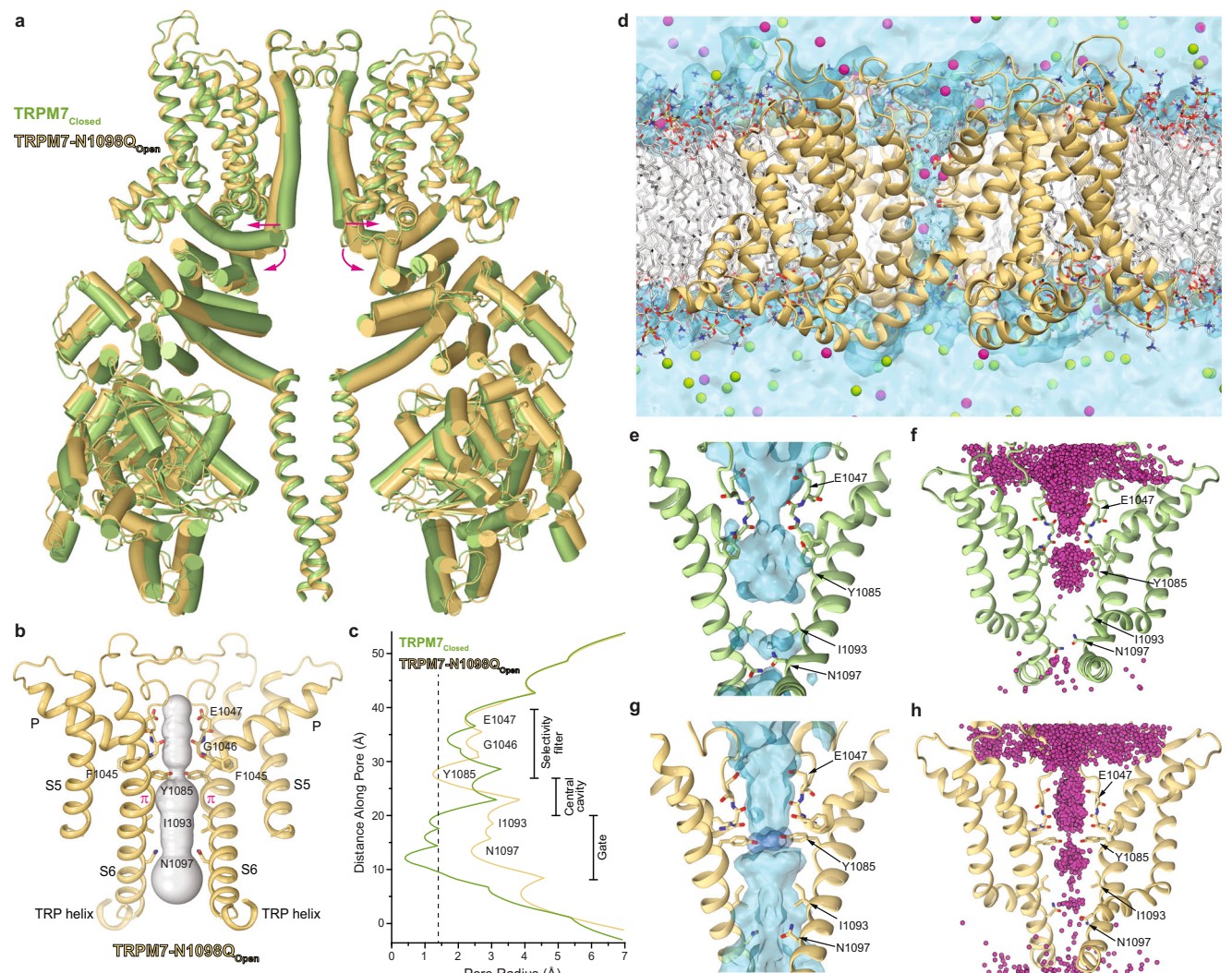

**Fig. 3 | Structure of TRPM7 with the gain-of-function mutation N1098Q.**
**a** Superposition of TRPM7$_{Closed}$ (green) and TRPM7-N1098Q$_{Open}$ struc-
tures. Only two of four subunits are shown, with the front and back subunits
omitted for clarity. Pink arrows show domain movements. **b** Pore-forming
domain of TRPM7-N1098Q$_{Open}$ with the residues contributing to the pore
lining shown as sticks. Only two of four subunits are shown, with the front and
back subunits omitted for clarity. The pore profile is shown as a space-filling
model (grey). The π-bulge in the middle of S6 is labeled. **c** Pore radius for
TRPM7-N1098Q$_{Open}$ (yellow) and TRPM7$_{Closed}$ (green) calculated using HOLE.
The vertical dashed line denotes the radius of a water molecule, 1.4 Å. **d** A
snapshot of the MD simulated system with TRPM7 shown as yellow ribbons,
lipid bilayer acyl chains in grey and hydrophilic

head groups as sticks, water as a light blue continuum, and Na$^+$ and Cl$^-$ ions as
magenta and green spheres, respectively. **e** Water occupancy of the TRPM7$_{Closed}$
channel (blue surface) in MD simulations with no applied voltage. Residues lining
the pore are shown in sticks. **f** Cumulative distribution of Na$^+$ ions (magenta
spheres) in the TRPM7$_{Closed}$ channel MD simulations with no applied voltage.
**g** Water occupancy of the TRPM7-N1098Q$_{Open}$ channel (blue surface) in MD
simulations with no applied voltage. The hydroxyl groups of Y1085 residues (shown
as dark blue surface) contribute to the permeation pathway. Residues lining the
pore are shown in sticks. **h** Cumulative distribution of K$^+$ ions (magenta spheres) in
the TRPM7-N1098Q$_{open}$ channel MD simulations under applied voltage.

pores in TRPM7-N1098Q and TRPM7$_{Closed}$. The narrow part of the gate
region (residues I1093 and N1097) in TRPM7-N1098Q appears to be
much wider than in TRPM7$_{Closed}$ (Fig. 3a, b). Measurements of the pore
radius (Fig. 3c) confirmed its increase in the gate region (N1097) from
<0.5 Å in TRPM7$_{Closed}$ to >2.3 Å in TRPM7-N1098Q. The narrowest
point of the pore (~1.4 Å) was formed by four tyrosines Y1085 (one per
subunit), which pointed their benzene ring-containing side chains
toward the pore center. Emphasizing the importance of these tyr-
osines for TRPM7 function, substitutions of Y1085 with either pheny-
lalanine or serine resulted in loss-of-function effects[47].

To further confirm that TRPM7-N1098Q represents an open
channel, we performed molecular dynamics (MD) simulations of
TRPM7-N1098Q in parallel with simulations of TRPM7$_{Closed}$ as a con-
trol. We set up a modeling system that included the TMD and the TRP
helix of the protein embedded into a POPC lipid bilayer solvated with

water and 150 mM NaCl (Fig. 3d, Supplementary Fig. 5 and Supple-
mentary Table 2). Our ~2 µs-long equilibrium MD simulations con-
firmed that the pore of TRPM7-N1098Q is permeable to water and
cations while neither water nor ions permeate through the
TRPM7$_{Closed}$ pore, which is sealed at the gate region by the side chains
of I1093 and N1097 (Fig. 3e–h and Supplementary Table 3).

In MD simulations of TRPM7-N1098Q in the absence of applied
voltage, cation permeation through the lower gate and the ring of
Y1085 residues was only occasional (Supplementary Movie 1). To fur-
ther validate channel conductance, we performed MD simulations of
TRPM7-N1098Q in KCl solution under applied voltage (600 mV) with
elevated ionic concentration (300 mM KCl) following the standard
practice in such simulations[50,51]. During the applied voltage simula-
tions, the channel showed continuous cation conductance (Supple-
mentary Fig. 6a–d, Supplementary Table 3, and Supplementary

Movie 2). The negatively charged chloride ions never entered the pore during any of the simulations confirming that TRPM7 is a channel with strong cation selectivity (Supplementary Fig. 6e–g). Hence, the structure of TRPM7-N1098Q in the apo condition represents an open state, TRPM7-N1098Q$_{open}$, in good agreement with our whole-cell electrophysiological analysis and single-channel recordings (Fig. 2a, b).

Furthermore, MD simulations showed that hydroxyl groups of Y1085 contribute to the permeation pathway of TRPM7-N1098Q$_{open}$ (Supplementary Fig. 6a–d). The side chain hydroxyl groups of Y1085 formed a hydrogen bonding network with water molecules in the channel. The lack of hydroxyl groups in Y1085F mutant or their larger separation in Y1085S mutant likely disturb this hydrogen bonding network, which in turn leads to improper hydration/dehydration of permeant cations, impedes their conductance through the pore and results in the loss-of-function effects observed for these mutants[47]. The planar, symmetrical arrangement of the Y1085 side chains in TRPM7-N1098Q$_{open}$, characterized by a stable $C_\alpha$–$C_\beta$ torsion angle ($\chi_1$) of nearly −180° was maintained throughout all MD simulations, except a single short simulation under high voltage. In contrast, the Y1085 side chains in TRPM7$_{Closed}$ fluctuated (between $\chi_1 = −80°$ and −180°) during all MD simulations.

To explore the mechanism of channel opening, we compared the S6-TRP helix region in TRPM7$_{Closed}$ and TRPM7-N1098Q$_{open}$. Upon channel opening, the S6 helices became two α-helical turns longer, while TRP helices became two α-helical turns shorter (Supplementary Fig. 6h) and rotated by ~6° in the plane of the membrane (Supplementary Fig. 6i). These rearrangements caused the substantial dilation of the ion channel pore and slight movement of S5 away from the pore center, whereas the rest of the channel remained virtually unaffected (Fig. 3a and Supplementary Movie 3). Such profound local conformational rearrangements in S6 and TRP helices during gating have not been reported for other TRPM channels[4,5] but were previously observed in TRPV3[49]. However, in contrast to TRPV3, where a local α-to-π transition in S6 accompanies the channel opening, the closed and open states of TRPM7 maintain the same helical conformation of S6, with the π-bulge in the middle (Fig. 3b and Supplementary Fig. 6h).

To understand how the N1098Q substitution stabilizes the open state, we compared the behavior of S6 and TRP helices during our ~2 μs-long MD simulations of TRPM7$_{Closed}$ and TRPM7-N1098Q$_{open}$. In TRPM7$_{Closed}$, the S6-TRP helix region was stabilized by a hydrogen bond between the N1098 side chain amide and the backbone carbonyl of A1094 (Supplementary Fig. 6j, k). Additional stability to this conformation was provided by the network of transient hydrogen bonds between the carboxamide groups of N1097 and N1098. In TRPM7-N1098Q$_{open}$, the side chains of N1097 and Q1098 were far apart and no longer interacted. Instead, the open conformation was stabilized by a new hydrogen bond between the N1097 side chain amide and the backbone carbonyl of I1093 (Supplementary Fig. 6l, m). Taken together, our results suggest that hydrogen bonds mediated by N1097 and N1098 control the constitutive activity of the TRPM7 channel.

## Activation of TRPM7 by agonist naltriben

To study agonist-induced activation of the channel, we employed a potent activator of TRPM7, naltriben (NTB)[52]. Whole-cell currents measured from HEK 293T cells expressing wild-type TRPM7 were dramatically increased in the presence of NTB (Fig. 2c). Likewise, a drastic increase in TRPM7 activity was observed for the purified protein reconstituted into lipid bilayers (Fig. 2d). Infrequent openings of TRPM7 in the absence of agonist ($P_o = 0.02 \pm 0.01$) changed to the sustained single-channel activity after adding 2.5 μM NTB ($P_o = 0.44 \pm 0.25$, $n = 3$; 18,900 events), clearly demonstrating the sensitivity of purified TRPM7 to NTB. Collectively, the results of our functional experiments supported the notion that NTB stabilizes TRPM7 in the open conducting state[34].

To explore the agonist-induced TRPM7 activation structurally, we subjected the purified, nanodisc-reconstituted TRPM7 protein to single-particle cryo-EM in the presence of 1 mM NTB. Our data analysis revealed two distinct classes of particles. The first class yielded a 2.17-Å structure in the open state, TRPM7$_{NTB-open}$ (Fig. 4a), with the pore conformation closely resembling the one in TRPM7-N1098Q$_{open}$ (Fig. 4b). Indeed, measurements of the pore radius demonstrated that the gate region in TRPM7$_{NTB-open}$ is much wider than in TRPM7$_{Closed}$ but has a similar size to the gate region in TRPM7-N1098Q$_{open}$ (Fig. 4c). The narrow constriction of the pore at the gate region in both TRPM7$_{NTB-open}$ and TRPM7-N1098Q$_{open}$ is formed by the side chains of four asparagines N1097, and there is a π-bulge in the middle of every S6 (Figs. 3b and 4b). Accordingly, our MD simulations demonstrated that the ion channel pore of TRPM7$_{NTB-open}$ conducts both water and cations in the presence or absence of applied voltage (Fig. 4f, g and Supplementary Table 3).

Four densities with the distinct shape of NTB molecule, one per monomer of the TRPM7$_{NTB-open}$ tetramer, revealed agonist-binding sites (Fig. 4d and Supplementary Fig. 4g). These agonist binding sites had not been identified in TRPM channels before. Each site is located at the intersubunit interface and formed by the MHR4/Pre-S1 region of one subunit and the loop connecting the MHR4 helices α21 and α22 of the neighboring subunit (Fig. 4d). To verify this new type of binding sites functionally, we mutated key residues contributing to NTB coordination and tested the activation of TRPM7 mutants by NTB using Ca$^{2+}$ uptake measurements. We found that compared to wild-type, mutant TRPM7-D670A, TRPM7-L671A, TRPM7-M741A, TRPM7-S744A, and TRPM7-N752K channels demonstrated substantially weakened activation by NTB (Fig. 4e, Supplementary Fig. 7a and Supplementary Table 4). Since the L671A and N752K mutations caused the strongest rightward shift of the NTB concentration dependence, we also examined the double mutant TRPM7-L671A/N752K, which displayed even stronger reduction in activation potency of NTB. In addition, we tested TRPM7-L671A and TRPM7-N752K using patch-clamp current recordings and found that both mutants reduced sensitivity of TRPM7 to 50 and 100 μM NTB (Supplementary Fig. 7b–d). Therefore, the results of our mutagenesis combined with functional recordings supported the idea that the agonist-induced activation of TRPM7 is mediated by binding of NTB to the sites identified in TRPM7$_{NTB-open}$.

We compared the TRPM7$_{NTB-open}$ and TRPM7$_{Closed}$ structures to highlight the molecular mechanism underlying the agonist-induced channel opening. As it is clearly illustrated by their superposition (Fig. 5a–d), NTB binding pulls the MHR4/Pre-S1 region of one subunit towards the MHR4 domain of the neighboring subunit. During this motion, the entire NTD of each individual subunit undergoes a ~21° rigid-body rotation. Such dramatic movement of the NTDs causes a ~21° rotation of the TRP helices, leading to the transformation of the S6-TRP helix connection (Supplementary Fig. 8a). Consequently, S6 elongates by two helical turns, the TRP helix becomes two helical turns shorter, and the pore dilates for ion and water conductance (Fig. 5e, f and Supplementary Movie 3).

The second class of particles revealed by the processing of cryo-EM data collected from TRPM7 in the presence of NTB yielded a 2.44-Å resolution closed-state structure TRPM7$_{NTB-Closed}$. This structure is very similar to TRPM7$_{Closed}$, with the pore sterically sealed at the narrow constriction by the side chains of I1093 and N1097 (Supplementary Fig. 8b, c). Inspection of the cryo-EM map revealed four identical densities, one per subunit of TRPM7$_{NTB-Closed}$ tetramer, with the characteristic shape of NTB (Supplementary Fig. 4g). Each of these densities unambiguously identified NTB binding sites located in the TMD region that faces the cytoplasmic leaflet of the membrane, at the interfaces between the S1–S4 and pore domains (Supplementary Fig. 8d), distally from the NTB binding sites in TRPM7$_{NTB-open}$ (Fig. 4d). The sites in TRPM7$_{NTB-Closed}$ are contributed by the N-terminal part of

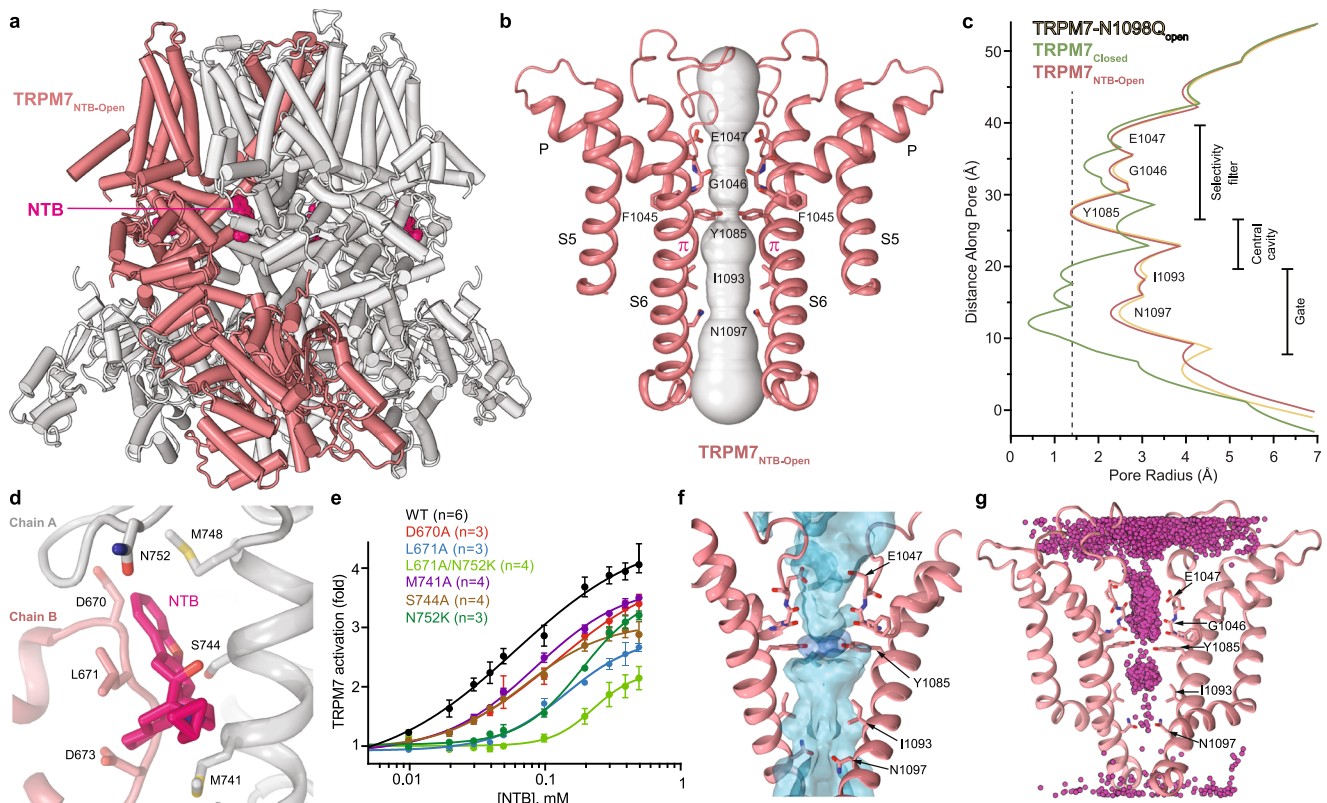

**Fig. 4 | Structure of TRPM7 in complex with agonist naltriben. a** Open-state structure TRPM7$_{NTB-open}$, with one subunit colored pink and the other three grey and molecules of NTB shown as space-filling models (bright pink). **b** Pore-forming domain of TRPM7$_{NTB-Open}$ with the residues contributing to pore lining shown as sticks. Only two of four subunits are shown, with the front and back subunits omitted for clarity. The pore profile is shown as a space-filling model (grey). The π-bulge in the middle of S6 is labeled. **c** Pore radius for TRPM7-N1098Q$_{open}$ (yellow), TRPM7$_{NTB-Open}$ (pink), and TRPM7$_{Closed}$ (green) calculated using HOLE. The vertical dashed line denotes the radius of a water molecule, 1.4 Å. **d** Close-up view of the NTB binding site. The molecule of NTB (bright pink) and residues involved in its binding are shown in sticks. **e** Concentration-dependences for activation of wild-

type (WT) and mutant TRPM7 channels by NTB, determined using the Ca$^{2+}$ influx assay as outlined in Supplementary Fig. 7a. Curves through the points (mean ± SEM) are the logistic equation fits; $n$, the number of independent measurements. The corresponding values of $IC_{50}$ and $n_{Hill}$ are provided in Supplementary Table 4. Source data are provided. **f** Water occupancy of the TRPM7$_{NTB-Open}$ channel (blue surface) in MD simulations with no applied voltage. The hydroxyl groups of the Y1085 residues that contribute to the permeation pathway are shown as dark blue surfaces. Residues lining the pore are shown in sticks. **g** Cumulative distribution of K$^+$ ions (magenta spheres) in TRPM7$_{NTB-Open}$ pore during MD simulations under applied voltage.

S3, C-terminal part of S4, S4–S5 linker and TRP helix (Supplementary Fig. 8d). Because the homologous site in TRPV channels is known as the vanilloid site[53], we will refer to the site in TRPM7 as vanilloid-like. Since the TRPM7$_{NTB-Closed}$ structure is nearly identical to the apo-state TRPM7$_{Closed}$ structure, the vanilloid-like site in TRPM7 does not serve as an activating binding site. Notably, a comparison of NTB molecules from TRPM7$_{NTB-open}$ and TRPM7$_{NTB-closed}$ structures shows that they are sterically identical and do not represent optical isomers (enantiomers) (Supplementary Fig. 8f–h).

### Inhibition of TRPM7 by potent selective antagonists VER155008 and NS8593

To study the structural mechanisms of TRPM7 inhibition and identify antagonist-binding sites, we chose two highly potent small-molecule inhibitors, VER155008 (VER)[40] and NS8593 (NS)[54], that have very different chemical structures from the agonist NTB (Fig. 2f). In the presence of either one of these inhibitors, the amplitude of TRPM7-mediated currents was greatly reduced (Fig. 2e). Using Ca$^{2+}$ imaging we determined the half-maximal inhibitory concentration ($IC_{50}$) of 0.23 ± 0.03 μM ($n = 6$) for VER and 0.91 ± 0.08 μM ($n = 5$) for NS (Supplementary Fig. 10 and Supplementary Table 4). To identify the antagonist-binding site location, we solved the structure of TRPM7 in complex with VER. We purified the TRPM7 protein and subjected it to single-particle cryo-EM in the presence of 100 μM VER. Cryo-EM data

analysis revealed a single population of particles that yielded a closed-state structure of TRPM7 bound to VER, TRPM7$_{VER-closed}$ (Fig. 6a). The TRPM7$_{VER-closed}$ structure had a narrow pore, very similar to the pore in the apo-state TRPM7$_{Closed}$ structure (Fig. 6b, c). However, in stark contrast to TRPM7$_{Closed}$, identical densities with the distinct shape of VER molecule (Supplementary Fig. 4h) were found in four (one per subunit) vanilloid-like sites in TRPM7$_{VER-closed}$ (Fig. 6d).

To confirm that it is VER binding to the vanilloid-like site that causes inhibition of TRPM7-mediated currents (Fig. 2e), we tested the ability of VER to trigger the closure of the TRPM7-N1098Q channel, structurally characterized as the open channel in the absence of ligands (Fig. 3). First, we conducted functional analysis and confirmed that TRPM7-N1098Q retained sensitivity to VER and NS, albeit the potency of both inhibitors was substantially reduced (Supplementary Fig. 10 and Supplementary Table 4). Similar to the ligand-free apo condition, the purified TRPM7-N1098Q protein subjected to cryo-EM in the presence of 100 μM VER was represented by a single class of particles. This time, however, the corresponding cryo-EM reconstruction yielded a 2.99-Å resolution structure in the closed state, TRPM7-N1098Q$_{VER-Closed}$, where the pore appeared to have a nearly identical size of TRPM7$_{Closed}$ and TRPM7$_{VER-closed}$ (Fig. 6c). Importantly, we found VER in TRPM7-N1098Q$_{VER-closed}$ bound to the same vanilloid-like sites as in TRPM7$_{VER-closed}$, strongly supporting the idea that these are the sites that mediate TRPM7 inhibition by VER. Similar

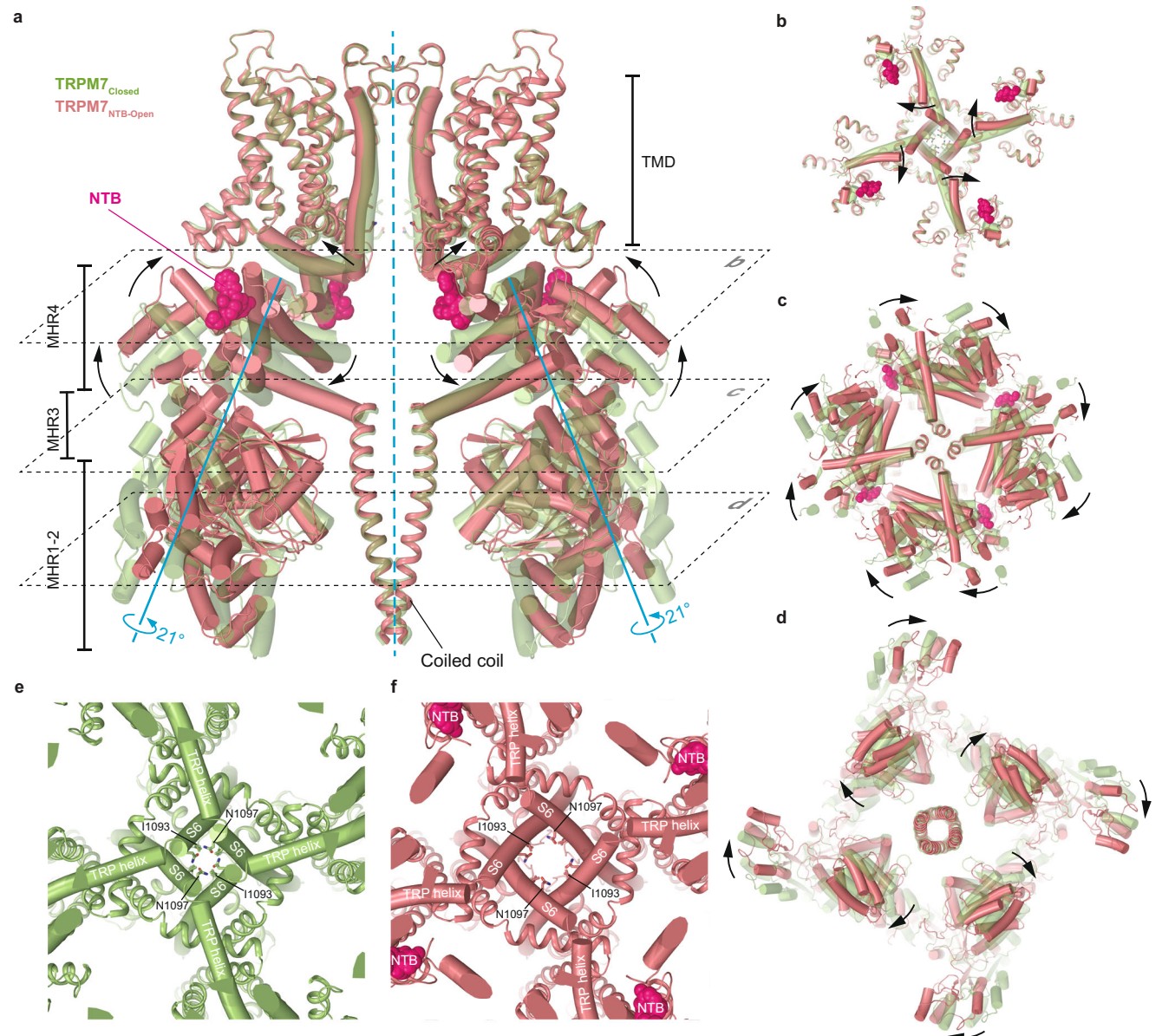

**Fig. 5 | Mechanism of the agonist-induced opening of TRPM7. a** Superposition of TRPM7$_{NTB-Open}$ (red) and TRPM7$_{Closed}$ (green) structures viewed parallel to the membrane. NTB molecules are shown as space-filling models (bright pink). Only two of four subunits are shown, with the front and back subunits omitted for clarity. Rotation axes are shown in blue, while arrows depict domain movements during channel opening. **b–d** Sections of TRPM7$_{NTB-open}$ (red) and TRPM7$_{Closed}$ (green) superposition indicated by dashed lines in (**a**), viewed perpendicular to the membrane, along the axis of fourfold rotational symmetry. Domain movements during NTB-induced channel opening are indicated by the black arrows. **e, f** Close-up view of the intracellular pore entrance in TRPM7$_{Closed}$ (**e**) and TRPM7$_{NTB-Open}$ (**f**) viewed along the axis of the 4-fold rotational symmetry. Residues forming the narrow constriction at the gate region of the pore are shown as sticks. NTB molecules are shown as space-filling models (bright pink).

cryo-EM experiments with 300 μM NS resulted in a 2.92-Å resolution closed-state TRPM7-N1098Q$_{NS-Closed}$ structure, where NS was also found bound to the vanilloid-like sites (Fig. 6f and Supplementary Figs. 4i and 8e).

We also confirmed that inhibition of TRPM7 is mediated by the vanilloid-like sites by mutating residues that contribute to VER and NS binding and comparing inhibition of Ca$^{2+}$ influx in mutant and wild-type channels (Fig. 6e, g). Thus, compared to wild-type channels, the potency of TRPM7-W1111A inhibition by VER was reduced ~500 times (Fig. 6e, Supplementary Fig. 10a, c and Supplementary Table 4). Weakening of VER inhibition by A981L, A981V and M991A substitutions was so strong that it was not feasible to calculate the $IC_{50}$ value in the working range of VER concentrations (Fig. 6e). Less dramatic compared to VER, but substantial weakening of TRPM7 inhibition by

mutations in the vanilloid-like site was observed for NS (Fig. 6g and Supplementary Table 4). Among all tested mutations, the strongest effects on TRPM7 inhibition by VER and NS were observed for substitutions of A981 with leucine or valine (Fig. 6e, g). It appears that the small side chain of alanine is critical for the high-affinity binding of VER and NS. Correspondingly, a bulky side chain introduced at this position sterically occludes the site and prevents the inhibitor binding. Overall, functional characterization of the mutant TRPM7 channels verifies our VER- and NS-bound structures and supports the idea that the vanilloid-like site serves as the primary inhibitory site in TRPM7.

To provide further evidence that VER and NS use the same vanilloid-like site to inhibit TRPM7 activity, we performed experiments with the TRPM7-M991A mutant, which is not sensitive to 1–10 μM VER (Fig. 6e) but presumably retains the ability to bind this ligand. Indeed,

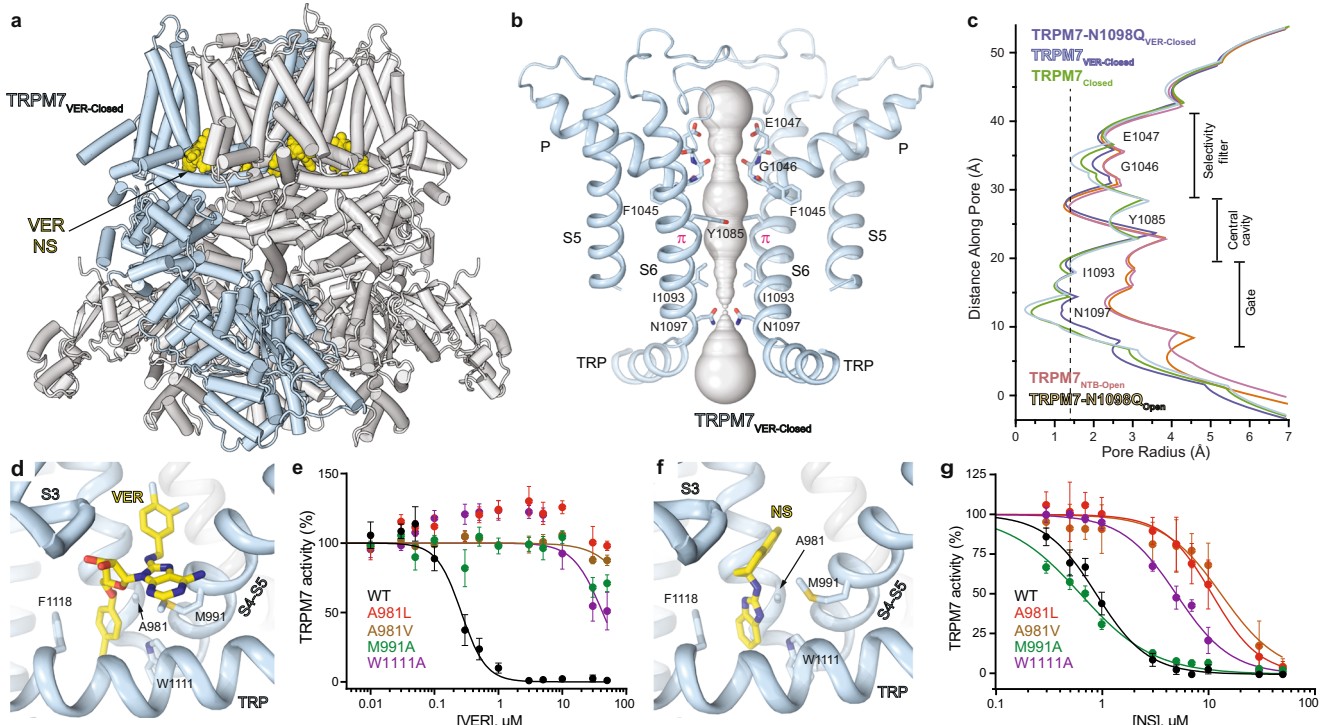

**Fig. 6 | Structures of TRPM7 in complex with inhibitors VER155008 and NS8593. a** Structure of TRPM7$_{VER-Closed}$, with one subunit colored blue and the other three grey and molecules of VER155008 shown as space-filling models (yellow). **b** Pore-forming domain in TRPM7$_{VER-Closed}$ with the residues contributing to pore lining shown as sticks. Only two of four subunits are shown, with the front and back subunits omitted for clarity. The pore profile is shown as a space-filling model (grey). The π-bulge in the middle of S6 is labeled. **c** Pore radius for TRPM7$_{Closed}$ and TRPM7 in complex with inhibitors compared to the pore radius for the open-state structures, all calculated using HOLE. The vertical dashed line denotes the radius of

a water molecule, 1.4 Å. **d, f** Close-up view of the vanilloid-like site in TRPM7$_{VER-Closed}$ (**d**) and TRPM7-N1098Q$_{NS-Closed}$ (**f**), with the inhibitor molecule (yellow) and residues contributing to its binding shown in sticks . **e, g** Concentration-dependencies for inhibition of wild-type (WT) and mutant TRPM7 by VER155008 (**e**) or NS8593 (**g**) assessed using the Ca$^{2+}$ influx assay. Curves through the points (mean ± SEM) are logistic equation fits. *n*, the number of independent measurements. The values of $IC_{50}$ and $n_{Hill}$ are provided in Supplementary Table 4. Source data are provided.

the TRPM7-M991A mutant was more sensitive to NS in the absence of added VER ($IC_{50}$ = 0.59 ± 0.05 μM, *n* = 5) than in the presence of 10 μM VER ($IC_{50}$ 1.28 ± 0.09 μM, *n* = 5) (Supplementary Fig. 9), confirming that these two inhibitors compete for the same binding site. Finally, we inspected TRPM7$_{Closed}$ and found that the vanilloid-like site in this apo-state structure is occupied by an acyl tail of a lipid molecule that protrudes from the base of the S1–S4 bundle (Supplementary Fig. 8e). Accordingly, VER and NS appear to stabilize the closed conformation of TRPM7 by substituting an endogenous lipid in the vanilloid-like binding pocket.

## Discussion
We used cryo-EM to solve structures of TRPM7 in the resting (apo) closed state, open by the gain-of-function mutation N1098Q or by the agonist NTB states, and closed states inhibited by the antagonists VER155008 or NS8593 (Fig. 7). Our closed-state structures represent the same conformation of the channel as the previously published structures of TRPM7[42] but include more molecular details because of their higher resolution. For example, our structures reveal the TRPM7 regions that have not been resolved before (Supplementary Fig. 4c). In addition, well-resolved densities around the TMD of TRPM7 allowed us to model many more annular lipids, providing a guide for studying their regulatory roles.

When compared to the closed-state structures, our open-state structures demonstrate dramatic conformational rearrangements that occur during TRPM7 activation. Thus, the N-terminal domains of the neighboring subunits, which are uncoupled in the absence of agonist, are tied and pooled to each other upon binding of NTB at the

intersubunit interfaces. As a result, concerted rigid-body movements of the individual subunits produce iris-like transformations of the cytoplasmic domain that culminate in ion channel opening (Figs. 5 and 7 and Supplementary Movie 3). This structural mechanism of ligand-induced opening is different from those revealed by the previously solved structures of the first TRPM group (TRPM2/4/5/8) representatives TRPM2[4,7,8], TRPM5[5], and TRPM8[6]. While the open-state structures remain to be seen for other representatives of the second group of TRPM channels (TRPM1/3/6/7), they are likely to have the mechanism of agonist-induced activation exemplified by TRPM7 in the present study.

Compared to the agonist-induced activation, different conformational rearrangements accompany TRPM7 opening caused by the gain-of-function mutation N1098Q (Fig. 3). Here, all structural changes in TRPM7-N1098Q are confined to the pore region, with the cytosolic domains left unaffected (Fig. 3a). Compared to the remote allosteric effect of the agonist NTB, the N1098Q-induced channel opening appears to result from the fine-tuning of open-closed state equilibrium by directly altering the elements of the TRPM7 gating machinery. Our MD simulations support this hypothesis and highlight specific interactions between the residues at the intracellular pore entrance (N1097 and N1098) that stabilize the closed-pore conformation in TRPM7 and open-pore conformation in TRPM7-N1098Q$_{open}$ (Supplementary Fig. 6j–m). This mechanism of channel opening by the gain-of-function mutation uncovers the previously unknown structural basis of constitutive activity in TRP channels that can serve as a template for the analysis of disease-causing mutations in *TRPM7*[35–39], *TRPM3*[55], and other TRP channels.

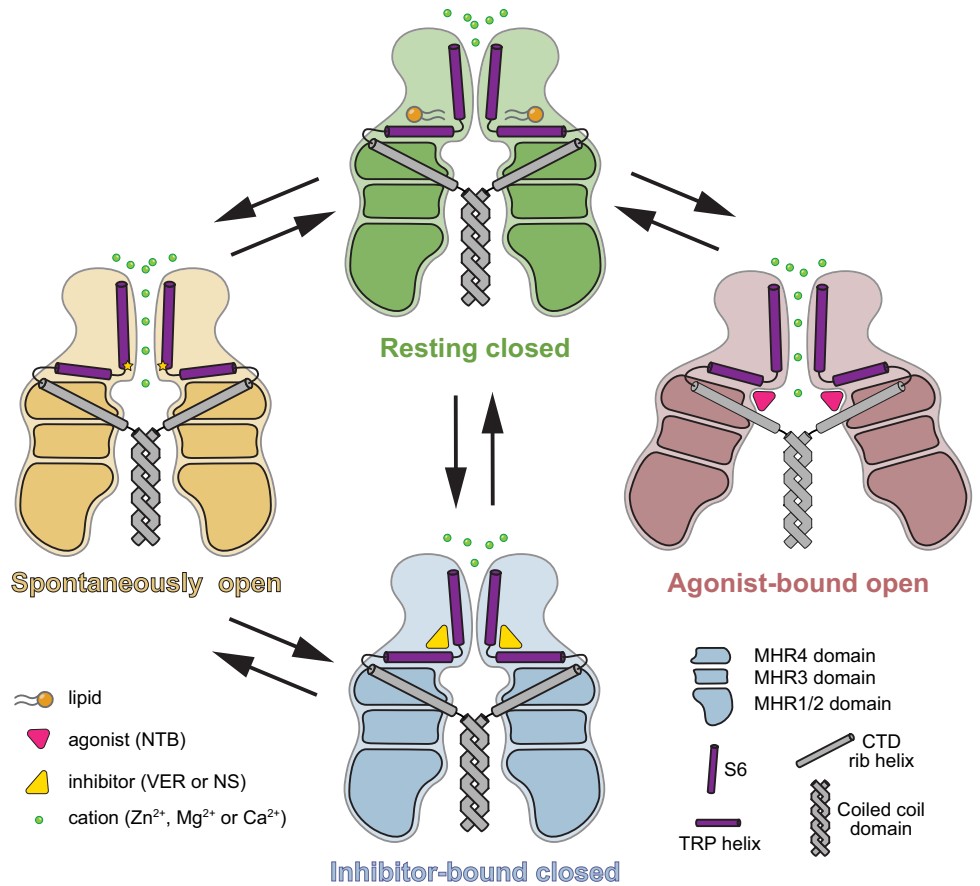

**Fig. 7 | Mechanisms of TRPM7 activation and inhibition.** Schematic representation of conformational changes in TRPM7 during activation and inhibition. Upon spontaneous activation, the conformational changes are confined within the TMD. Conversely, binding of the agonist NTB induces substantial conformational rearrangements in both the N-terminal MHR1-4 domains and the TMD, leading to the channel opening. The inhibitor binding locks TRPM7 in the closed conformation resembling the resting apo state.

Which one of the two open-state structures, TRPM7-N1098Q$_{open}$ or TRPM7$_{NTB-open}$, represents the physiological state of TRPM7 in the cellular membrane? We hypothesize that TRPM7-N1098Q$_{open}$ represents the spontaneously open state of the channel because it does not require energetically costly rearrangement of the MHR regions. This state may occur upon removal of intracellular $Mg^{2+}$, which likely inhibits TRPM7 through interaction with the adjacent to N1098 asparagine N1097[47]. On the other hand, similar to TRPM2[4,7], TRPM7 activity can be regulated by endogenous ligands acting by binding to the intracellular domains. In this case, TRPM7 can adapt the conformation represented by the TRPM7$_{NTB-open}$ structure. Further experiments are needed to address this intriguing question.

The present study also discovers the mechanism of TRPM7 inhibition by two potent and structurally unrelated antagonists, VER155008 and NS8593. They share the same binding site contributed by the N-terminal part of S3, C-terminal part of S4, S4–S5 linker and TRP helix. This site, often called the vanilloid site in other TRP channels, has not been previously identified in TRPM7 but was shown to bind the antagonist NDNA in TRPM5[5], activator GNE551 in TRPA1[56], and a variety of ligands and lipids in TRPV channels[53]. Due to the similarity of the inhibitor-bound and resting states (Fig. 6c), the mechanism of TRPM7 inhibition by VER155008 and NS8593 appears to be the stabilization of the closed state (Fig. 7).

Collectively, our study reveals the molecular basis of TRPM7 gating and inhibition and provides mechanistic insights into the function of this ion channel, which is essential for the cellular uptake of divalent cations. Selective pharmacological modulation of TRPM7 was proposed to be beneficial for patients with immune and cardiovascular disorders, tumors and other pathologies[11–13]. Accordingly, our structures, including the identified agonist and antagonist binding sites, can serve as templates for the design of pharmacological modulators of TRPM7.

## Methods

### Constructs

For structural experiments and planar lipid bilayer recordings, cDNA for mouse *Trpm7* (NM_021450) truncated C-terminally (the corresponding protein residues 1–1280)[42] was introduced into the *pEG-BacMam* vector for protein expression in mammalian cells[57], with an N-terminal region coding for the streptavidin affinity tag (residues WSHPQFEK), followed by the green fluorescent protein (GFP) and thrombin cleavage site (residues LVPRG), as described before[58,59]. For functional experiments, mouse *Trpm7* was introduced into the *pIRES2-eGFP* expression vector reported previously[36]. Point mutations in TRPM7 were introduced using the QuikChange system (Thermo Fisher Scientific) according to the manufacturer's protocol and verified by sequencing (Eurofins, Germany).

### Protein expression and purification

For cryo-EM studies, TRPM7 bacmids and baculoviruses were produced using the standard procedures[57]. Briefly, baculovirus was made in Sf9 cells (Thermo Fisher Scientific, mycoplasma test negative, GIBCO #12659017) for ~96 h and added to suspension-adapted HEK 293S cells lacking N-acetyl-glucosaminyltransferase I (GnTI⁻, mycoplasma test negative, ATCC #CRL-3022) that were maintained at 37 °C and 6% $CO_2$ in Freestyle 293 media (Gibco-Life Technologies #12338-

018) supplemented with 2% FBS. To reduce TRPM7 cytotoxicity, 10 μM ruthenium red was added to the suspension of HEK 293S cells. To enhance protein expression, sodium butyrate (10 mM) was added 12 h after transduction, and the temperature was reduced to 30 °C. The cells were harvested 48 h after transduction by 15-min centrifugation at $5471 \times g$ using a Sorvall Evolution RC centrifuge (Thermo Fisher Scientific). The cells were washed in the phosphate buffer saline (PBS) pH 8.0 and pelleted by centrifugation at $3202 \times g$ for 10 min using an Eppendorf 5810 centrifuge.

The cell pellet was resuspended in the ice-cold buffer containing 20 mM Tris pH 8.0, 150 mM NaCl, 0.8 μM aprotinin, 4.3 μM leupeptin, 2 μM pepstatin A, 1 μM phenylmethylsulfonyl fluoride (PMSF), and 1 mM β-mercaptoethanol (βME). The suspension was supplemented with 1% (w/v) glyco-diosgenin (GDN) or digitonin, and the cells were lysed at constant stirring for 2 h at 4 °C. Unbroken cells and cell debris were pelleted in the Eppendorf 5810 centrifuge at $3202 \times g$ and 4 °C for 10 min. Insoluble material was removed by ultracentrifugation for 1 h at $186,000 \times g$ in a Beckman Coulter centrifuge using a 45 Ti rotor. The supernatant was added to 5 ml of strep resin, which was then rotated for 20 min at 4 °C. The resin was washed with 10 column volumes of the wash buffer containing 20 mM Tris pH 8.0, 150 mM NaCl, 1 mM βME, and 0.01% (w/v) GDN, and the protein was eluted with 15 ml of the same buffer supplemented with 2.5 mM D-desthiobiotin. The eluted protein was concentrated to 0.5 ml using a 100-kDa NMWL centrifugal filter (MilliporeSigma™ Amicon™) and then centrifuged in a Sorvall MTX 150 Micro-Ultracentrifuge (Thermo Fisher Scientific) for 30 min at $66,000 \times g$ and 4 °C using a S100AT4 rotor before injecting it into a size-exclusion chromatography (SEC) column. The protein was purified using a Superose™ 6 10/300 GL SEC column attached to an AKTA FPLC (GE Healthcare) and equilibrated with the buffer containing 150 mM NaCl, 20 mM Tris pH 8.0, 1 mM βME, and 0.01% (w/v) GDN. The tetrameric peak fractions were pooled and concentrated to 2–3 mg/ml using the 100-kDa NMWL centrifugal filter.

For TRPM7 reconstitution into MSP2N2 nanodiscs, the purified TRPM7 protein was mixed with the purified MSP2N2 protein and lipids POPC:POPE:POPG (3:1:1, molar ratio; 1-palmitoyl-2-oleoyl-glycero-3-phosphocholine−POPC, 1-palmitoyl-2-oleoyl-glycero-3-phosphoethanolamine−POPE, 1-palmitoyl-2-oleoyl-glycero-3-phosphoglycerol−POPG; Avanti Polar Lipids, USA) at a molar ratio of 1:3:166 (monomer:MSP2N2:lipid). The MSP2N2 protein was stored in a buffer containing 150 mM NaCl and 20 mM Tris (pH 8.0). The lipids were resuspended to a concentration of 100 mg/ml in 150 mM NaCl, 20 mM Tris (pH 8.0) and subjected to 5–10 cycles of freezing in liquid nitrogen and thawing in a water bath sonicator. The nanodisc mixture (500 μl) was rocked at room temperature for 1 h. Subsequently, the nanodisc mixture was supplemented with 40 mg of Bio-beads SM2 (Bio-Rad) pre-wet in the buffer containing 20 mM Tris pH 8.0, 150 mM NaCl, and 1 mM βME and subjected to rotation at 4 °C. After 1 h of rotation, 40 mg more of Bio-beads SM2 was added, and the resulting mixture was rotated at 4 °C for another -14–20 h. The Bio-beads SM2 were then removed by pipetting, and TRPM7 reconstituted in nanodiscs was purified from empty nanodiscs by SEC using the Superose™ 6 10/300 GL column equilibrated in the buffer containing 150 mM NaCl, 20 mM Tris (pH 8.0), and 1 mM βME. The SEC fractions corresponding to TRPM7 reconstituted in nanodiscs were pooled and concentrated to 2.2–2.3 mg/ml using the 100-kDa NMWL centrifugal filter.

## Cryo-EM sample preparation and data collection
For the sample with naltriben, the nanodisc-reconstituted TRPM7 was supplemented with 200 μM of 1-(1,2R-dioctanoylphosphatidyl)inositol-4,5-bisphosphate (PIP$_2$), incubated at room temperature for 30 min, and then supplemented with 1 mM of naltriben (dissolved in DMSO) 5 min before grid freezing. For the samples with VER155008, 100 μM of VER155008 (dissolved in DMSO) was added to the protein, which was then incubated for 15 min at room temperature before grid

freezing. For the samples with NS8593, 300 μM of NS8593 (dissolved in DMSO) was added to the protein, which was then incubated for 15 min at room temperature before grid freezing. NS8593 was acquired from Tocris. VER155008 and naltriben were purchased from Sigma-Aldrich. Prior to sample application, CF 1.2/1.3, Au-50 (300-mesh) grids were plasma treated in a PELCO easiGlow glow discharge cleaning system (0.39 mBar, 15 mA, "glow" for 25 s, and "hold" for 10 s). A Mark IV Vitrobot (Thermo Fisher Scientific) set to 100% humidity and 4 °C was used to plunge-freeze the grids in liquid ethane after applying 3 μl of protein sample to their gold-coated side using the blot time of 3 s, blot force of 3, and wait time of 15 s. The grids were stored in liquid nitrogen before imaging.

Images of frozen-hydrated particles of TRPM7 in GDN (apo state) were collected in the Columbia University Cryo-EM Facility using the Leginon software[60] on a Titan Krios transmission electron microscope (TEM) (Thermo Fisher Scientific) operating at 300 kV and equipped with a post-column GIF Quantum energy filter and a Gatan K3 Summit direct electron detection (DED) camera (Gatan, Pleasanton, CA, USA). The total of 9,822 micrographs were collected in the counting mode with an image pixel size of 0.83 Å across the defocus range of −0.5 to −1.5 μm. The total dose of -58 e$^-$Å$^{-2}$ was attained by using the dose rate of -16 e$^-$pixel$^{-1}$s$^{-1}$ across 50 frames during the 2.5-s exposure time.

Images of frozen-hydrated particles of TRPM7-N1098Q in GDN were collected in the Columbia University Cryo-EM Facility using the Leginon software on a Titan Krios TEM operating at 300 kV and equipped with a post-column GIF Quantum energy filter and a Gatan K3 Summit DED camera. The total of 5521 micrographs were collected in the counting mode with an image pixel size of 0.83 Å across the defocus range of −0.5 to −1.5 μm. The total dose of -58 e$^-$Å$^{-2}$ was attained by using the dose rate of -16 e$^-$pixel$^{-1}$s$^{-1}$ across 50 frames during the 2.5-s exposure time.

Images of frozen-hydrated particles of TRPM7 in GDN with added 300 μM NS8593 were collected in the National Cryo-Electron Microscopy Facility (Frederick National Laboratory for Cancer Research) using the EPU software on a Titan Krios TEM operating at 300 kV and equipped with a post-column GIF Quantum energy filter and a Gatan K3 Summit DED camera. The total of 7659 micrographs were collected in the super resolution mode with an image pixel size of 0.428 Å across the defocus range of −0.5 to −1.5 μm. The total dose of -60 e$^-$Å$^{-2}$ was attained by using the dose rate of -31.57 e$^-$pixel$^{-1}$s$^{-1}$ across 50 frames during the 2.6-s exposure time.

Images of frozen-hydrated particles of TRPM7 in GDN with added 100 μM VER155008 were collected in the Columbia University Cryo-EM Facility using the Leginon software on a Titan Krios TEM operating at 300 kV and equipped with a post-column GIF Quantum energy filter and a Gatan K3 Summit DED camera. The total of 5738 micrographs were collected in the counting mode with an image pixel size of 0.83 Å across the defocus range of −0.5 to −1.5 μm. The total dose of -58 e$^-$Å$^{-2}$ was attained by using the dose rate of -16 e$^-$pixel$^{-1}$s$^{-1}$ across 50 frames during the 2.5-s exposure time.

Images of frozen-hydrated particles of TRPM7-N1098Q in GDN with added 100 μM VER155008 were collected in the Columbia University Cryo-EM Facility using the Leginon software on a Titan Krios TEM operating at 300 kV and equipped with a post-column GIF Quantum energy filter and a Gatan K3 Summit DED camera. The total of 4685 micrographs were collected in the counting mode with an image pixel size of 0.83 Å across the defocus range of −0.5 to −1.5 μm. The total dose of -58 e$^-$Å$^{-2}$ was attained by using the dose rate of -16 e$^-$pixel$^{-1}$s$^{-1}$ across 50 frames during the 2.5-s exposure time.

Images of frozen-hydrated particles of TRPM7 in MSP2N2 nanodiscs (apo state) were collected in the Columbia University Cryo-EM Facility using the Leginon software on a Titan Krios TEM operating at 300 kV and equipped with a post-column GIF Quantum energy filter and a Gatan K3 Summit DED camera. The total of 5087 micrographs were collected in the counting mode with an image pixel size of 0.83 Å

across the defocus range of −0.5 to −1.5 μm. The total dose of ~58 e⁻Å⁻² was attained by using the dose rate of ~16 e⁻ pixel⁻¹s⁻¹ across 50 frames during the 2.5-s exposure time.

Images of frozen-hydrated particles of TRPM7 in MSP2N2 nanodiscs with added 200 μM PIP$_2$ and 1 mM NTB were collected in the Columbia University Cryo-EM Facility using the Leginon software on a Titan Krios TEM operating at 300 kV and equipped with a post-column GIF Quantum energy filter and a Gatan K3 Summit DED camera. The total of 5738 micrographs were collected in the counting mode with an image pixel size of 0.83 Å across the defocus range of −0.5 to −1.5 μm. The total dose of ~58 e⁻Å⁻² was attained by using the dose rate of ~16 e⁻pixel⁻¹s⁻¹ across 50 frames during the 2.5-s exposure time.

## Image processing and 3D reconstruction

Single-particle cryo-EM data was processed in cryoSPARC 3.3.2[61] and Relion 4.0[62]. Movie frames were aligned using the patch motion correction in cryoSPARC 3.3.2 or MotionCor2 algorithm implemented in Relion 4.0. The contrast transfer function (CTF) estimation was performed using the patch CTF estimation. Following CTF estimation, micrographs were manually inspected and those with outliers in defocus values, ice thickness, and astigmatism as well as micrographs with lower predicted CTF-correlated resolution (higher than 6 Å) were excluded from further processing (individually assessed for each parameter relative to the overall distribution). Similar processing was used for all datasets. For example, for TRPM7 reconstituted in MSP2N2 nanodiscs (apo state), the total number of 1,336,887 particles were picked using internally generated 2D templates and extracted with the 300-pixel box size and then binned to the 128-pixel box size. After several rounds of reference-free 2D classifications and Heterogeneous Refinements in cryoSPARC 3.3.2 with one reference class and three automatically generated "garbage" classes, the best 579,518 particles were imported into Relion and re-extracted with the 300-pixel box size without binning. These particles were subjected to one round of 3D classification into nine classes without imposing symmetry restraints (C1 symmetry). At this point, two clusters of well-defined classes were observed, one with a folded coiled-coil domain and another with a completely disordered coiled-coil region. Particles representing four best classes with the folded coil-coiled region were combined, refined together (C1) and subjected to CTF refinements to correct for the beam-tilt, higher order aberrations, anisotropic magnification, per particle defocus, and per micrograph astigmatism[63]. The CTF-refined particles were subjected to Bayesian polishing and CTF refined again using the same procedure as described above. Polished and CTF-refined particles were 3D classified into ten classes (C1), and particles for the best two classes were imported into cryoSPARC 3.3.2. The final set of 178,432 particles representing the best classes was subjected to homogenous, non-uniform, and CTF refinements with C4 rotational symmetry. The reported resolution of 2.19 Å for the final map was estimated using the gold standard Fourier shell correlation (GSFSC). To improve the quality of reconstruction for the N-terminal region that experienced rigid-body movements, the final set of particles was C4 symmetry expanded (689,728 particles), and the N-terminal domain region was extracted from 2D images of individual particles by subtraction of other regions. The subtracted particles comprising only the N-terminal region were subjected to local refinement with a mask around this region and then to 3D classification without alignment in cryoSPARC 3.3.2. The final set of 325,415 subtracted particles representing the best classes with the resolved secondary structure was subjected to the local refinement with the same mask and yielded a map with the resolution of 2.62 Å (Supplementary Fig. 2j). For model building, we also created a composite map from the maps for the entire channel and the N-terminal region using the Combine-focused-maps package implemented in Phenix. The local resolution was calculated for the maps of the entire channel and the N-terminal region

separately using the FSC = 0.5 criterion. Cryo-EM density was visualized using UCSF ChimeraX[64].

## Model building

The core of the TRPM7$_{Closed}$ model was built in Coot[65] using the previously published cryo-EM structure of TRPM7 (PDB ID: 5ZX5)[42] as a guide. Regions that were not present in the previously published structure were built de novo, using the cryo-EM density as a guide. Other TRPM7 structures were built using TRPM7$_{Closed}$ as a guide. The models were tested for overfitting by shifting their coordinates by 0.5 Å (using Shake) in Phenix[66], refining each shaken model against the corresponding unfiltered half map, and generating densities from the resulting models in UCSF ChimeraX. The resulting models were real-space refined in Phenix 1.18 and visualized using UCSF ChimeraX and PyMOL (The PyMOL Molecular Graphics System, Version 2.0 Schrödinger, LLC.). The pore radius was calculated using HOLE[67].

## Aequorin-based Ca²⁺ influx assay

Measurements of [Ca²⁺]$_i$ were performed as reported previously with several modifications[2,52,54]. HEK 293T cells (mycoplasma test negative, ATCC #CRL3216) were maintained at 37 °C and 5% CO$_2$ in DMEM supplemented with 10% fetal calf serum, 100 μg/ml streptomycin and 100 U/ml penicillin (all from Thermo Fisher Scientific). Cells cultured in six-well plates (~60% confluence) were transfected with 2 μg/dish *Trpm7* plasmid DNA and 0.1 μg/dish *pGSA* plasmid DNA encoding eGFP fused to *Aequorea victoria* aequorin, using Lipofectamine 2000 (Thermo Fisher Scientific). Twenty-four hours after transfection, the cells were washed with Mg²⁺-free HEPES-buffered saline (Mg²⁺-free HBS) containing 140 mM NaCl, 6 mM KCl, 0.5 mM CaCl$_2$, 10 mM HEPES (pH 7.4) and 5 mM glucose, and mechanically resuspended in the Mg²⁺-free HBS. For reconstitution of aequorin, cell suspensions were incubated with 5 μM coelenterazine (Biaffin GmbH) in the Mg²⁺-free HBS for 30 min at room temperature. Cells were washed twice by centrifugation at 600 rpm for 5 min (Heraeus Pico 17 microcentrifuge, Thermo Fisher Scientific), resuspended in the Mg²⁺-free HBS and aliquoted into 96-well plates (1 × 10⁵ cells per well). Luminescence was detected at room temperature using a FLUOstar OPTIMA microplate reader (BMG LABTECH GmbH). To monitor the NS8593- and VER155008-induced inhibition of TRPM7-mediated Ca²⁺ influx, the extracellular concentration of Ca²⁺ was increased to 5 mM by injecting the CaCl$_2$-containing Mg²⁺-free HBS in the absence or presence of the inhibitors. The background Ca²⁺ influx was determined using cells transfected with an inactive mutant channel *Trpm7-P1040R*[68]. To study activation by naltriben, TRPM7-transfected cells were exposed to Mg²⁺-free HBS containing different concentrations of the agonist. The experiments were terminated by lysing cells with 0.1% (v/v) Triton X-100 in the Mg²⁺-free HBS to record the total bioluminescence. The bioluminescence rates (counts/s) were analyzed at 1-s intervals and calibrated as [Ca²⁺]$_i$ values using the following equation:

$$p[Ca^{2+}]_i = 0.332588\,(-log(k)) + 5.5593 \qquad (1)$$

where $k$ represents the rate of aequorin consumption, i.e., counts/s divided by the total number of counts.

To determine the $IC_{50}$ values for NS8593 and VER155008 ($EC_{50}$ for naltriben), the data were fitted with the following logistic equation:

$$E(c) = E_{min} + \frac{E_{max} + E_{min}}{1 + \frac{c^h}{IC_{50}^{\ h}}} \qquad (2)$$

where $E(c)$ is the functional effect at the concentration $c$ of the compound, $E_{min}$ is the minimal effect, $E_{max}$ is the maximal effect, $IC_{50}$ ($EC_{50}$) is the concentration of the compound that produces the half-maximal effect, and $h$ is the Hill coefficient. Fitting of dose-response curves was

performed using GraphPad Prism 8.4.0. Data are presented as the mean ± SEM.

## Patch-clamp measurements

Patch-clamp experiments were performed as reported previously, with a few modifications[69,70]. HEK 293T cells grown in 35-mm dishes to ~60% confluence were transiently transfected with *Trpm7* cDNAs in the *pIRES2-EGFP* vector (2 μg/dish). Patch-clamp experiments were conducted 18–22 h after transfection with cells displaying EGFP fluorescence. Whole-cell currents were recorded using an EPC10 patch-clamp amplifier and PatchMaster software (Version V2x69, Harvard Bioscience). Voltages were corrected for a liquid junction potential of 10 mV. Currents were elicited by voltage ramps from −100 mV to +100 mV over 50 ms acquired at 0.5 Hz. The inward and outward current amplitudes were measured at −80 mV and +80 mV and were normalized to the cell size as pA/pF. The capacitance was measured using the automated capacitance cancellation function of EPC10. The standard extracellular solution contained 140 mM NaCl, 2.8 mM KCl, 1 mM $CaCl_2$, 2 mM $MgCl_2$, 10 mM HEPES-NaOH, and 11 mM glucose (all from Sigma-Aldrich). Solutions were adjusted to pH 7.2 using an FE20 pH meter (Mettler Toledo) and to 290 mOsm using a Vapro 5520 osmometer (Wescor Inc). Patch pipettes were made of borosilicate glass (Science Products) and had a resistance of 2.0–3.7 MΩ when filled with the standard intracellular pipette solution containing 120 mM Cs-glutamate, 8 mM NaCl, 10 mM Cs-EGTA, 5 mM Cs-EDTA, and 10 mM HEPES-CsOH. For the experiment with naltriben, the intracellular solution contained 120 mM Cs-glutamate, 8 mM NaCl, 10 mM Cs-EGTA, 1.5 mM $MgCl_2$, and 10 mM HEPES-CsOH. The intracellular solution was also adjusted to pH 7.2 and 290 mOsm. Data are presented as the mean ± SEM Unless indicated otherwise, data were compared by unpaired *t*-test (GraphPad Prism 8.4.0). For multiple comparisons, ordinary one-way ANOVA (GraphPad Prism 8.4.0) was used. Significance was accepted at $P \le 0.05$.

## Planar lipid bilayer recordings

Planar lipid bilayers measurements were performed as described previously[51]. Briefly, planar lipid bilayers were formed from a 30 mM solution of synthetic lipid mix POPC:POPE:POPG at a 3:1:1 ratio (Anatrace, P516:P416:P616) in *n*-decane (Sigma-Aldrich). The solution was used to paint a bilayer in an aperture of ~150 μm diameter in a Meca chip (Nanion). Each cavity in the chip contained an individual integrated Ag/AgCl-microelectrode. The bathing solution contained 150 mM KCl, 0.02 mM $MgCl_2$, 1 μM $CaCl_2$ and 20 mM HEPES (pH 7.2). All the reagents (Sigma-Aldrich) were ultrapure (>99%). The bilayer capacitance was in the range of 5–10 pF.

The purified protein (10 ng/ml) was added to the bilayer-forming lipid mix (1 volume of protein to 1 volume of the lipid mix) and incubated for 30 min at 30 °C. After the bilayers had been formed by painting on the Meca chip, they did not show any single-channel activity. Only after the incubated protein-lipid mix was added by painting, the unitary currents were recorded using an Orbit mini device (Nanion) controlled by the pCLAMP 10.2 software (Molecular Devices). Data were low-pass filtered at 20 kHz and digitized at 1.22 kHz. Single-channel conductance events, all-points histograms, and other parameters were identified and analyzed using the Clampfit 10.3 software (Molecular Devices). Only recordings with one channel incorporated into the lipid bilayer were subjected to single-channel analysis. All recordings with more than one channel incorporated into lipid bilayer were discarded from the analysis. All experiments were performed at room temperature. All data were presented as mean ± SEM.

## System setup for molecular dynamics simulations

The TRPM7$_{Closed}$, TRPM7-N1098Q$_{open}$ and TRPM7$_{NTB-open}$ structures were used as initial atomic coordinates for MD simulations. Each structure was truncated to include residues 843 to 1147 (the TMD and TRP

helix) of the protein. Each truncated protein was inserted into a POPC bilayer and solvated with TIP3P water molecules and 150 mM NaCl using the CHARMM-GUI membrane builder[71,72]. The systems were set up for MD simulations using the tleap module of the AmberTools20 package[73]. Amber FF99SB-ILDN[74] and Lipid14[75] force field parameters were used for protein and lipids, respectively. The final systems contained 178,416–178,680 atoms, including 1220 protein residues, 411 lipid molecules, 34,262–34,354 water molecules, 96 $Na^+$ ions, and 116 $Cl^-$ ions.

## MD simulation without applied voltage

All MD simulations were carried out using the pmemd.cuda program of the Amber20 molecular dynamics package[73]. All equilibration and production simulations were carried out in NPT ensemble at 300 K temperature and 1 bar pressure with anisotropic scaling. A Langevin thermostat with a collision frequency of 1 $ps^{-1}$ and Berendsen barostat with a relaxation time of 1 ps were used as implemented in Amber20. All covalent bonds involving hydrogen atoms were constrained using the SHAKE algorithm[76], with the integration time step of 2 fs. Electrostatic interactions were calculated using the Particle mesh Ewald method[77], with a non-bonded interaction cutoff radius of 10 Å. Periodic boundary conditions were applied in all directions.

Each system was minimized and heated from 0.1 K to 100 K at the constant volume and from 100 to 300 K at the constant pressure, with all protein main-chain heavy atoms harmonically restrained at their original positions with the force constant of 20 kcal $mol^{-1}$ Å. The systems were then equilibrated for 160 ns, gradually releasing the restraints on the protein. To limit excessive fluctuations of the protein due to truncation, the $C_\alpha$ atoms in the terminal regions (residues 843–850 and 1134–1147) were restrained with the force constant of 5 kcal $mol^{-1}$ $Å^{-1}$ throughout production simulations. For TRPM7$_{Closed}$ and TRPM7-N1098Q$_{open}$ models, the backbone N-O distances of the TRP helix residues 1109–1123 were additionally restrained between 2.60 and 4.25 Å. For the TRPM7$_{NTB-open}$ model, the $C_\alpha$ atoms of residues 1108 to 1123 of the TRP helix and residues 913–921 of the S2–S3 linker were positionally restrained with the force constant of 5 kcal $mol^{-1}$ $Å^{-1}$. All other restraints were removed for the production runs. Production simulations were carried out for 0.8–1 μs.

## MD simulations under applied voltage

Equilibrated TRPM7-N1098Q$_{open}$ and TRPM7$_{NTB-open}$ were used as starting structures for ion conduction simulations. $Na^+$ ions were replaced with $K^+$ ions to improve simulated conductance, and additional $K^+$ and $Cl^-$ ions were added to the bulk solution to bring the KCl concentration up to 300 mM. The force field parameters and the simulation settings were the same as described in the "MD simulations without an applied voltage" section, unless otherwise specified. The initial systems were minimized and equilibrated in NPT ensemble for 20 ns, with all protein $C_\alpha$ atoms restrained with the force constant of 1 kcal $mol^{-1}$ Å. The $C_\alpha$ atom restraints were removed, and the systems were equilibrated for further 5 ns in NVT ensemble followed by production runs in NVT under applied electric field for 300–400 ns. All other restraints were set as described in the "MD simulations without an applied voltage" section. An external electric field was applied perpendicular to the membrane (along the Z direction) (using the "efz" option in Amber) to achieve the voltage of 600 mV across the membrane.

## Trajectory analysis

Post-processing and analysis of the trajectories were carried out using CPPTRAJ[78] module of AmberTools20 and VMD 1.9.4[79]. VMD 1.9.4 was used to visualize trajectories and generate molecular graphics.

## Channel water density and water permeation

Channel water permeation and occupancy (aligned around the origin at the center of mass of Y1085) were calculated at −20 Å < Z < 20 Å. For

all systems, the average water permeation per nanosecond was calculated as counts of the total downward ($-Z$) and upward ($+Z$) water permeation divided by ~60-ns blocks (each block was selected by skipping production run trajectories at every 60 ns for all systems) of AMBER production trajectories (Supplementary Table 2). Each successful count represented a downward ($-Z$) and upward ($+Z$) passing water, with the entry point of $Z = -2$ Å and the exit point of $Z = -18$ Å, whereby all snapshots were aligned with the center of mass at the Y1085 origin. Rapid movements of water were captured at 10-ps time frames, which is also the frequency of saving trajectory snapshots in our simulations.

## Ion channel conductance

The conductance of the TRPM7-N1098Q$_{open}$ and TRPM7$_{NTB-open}$ channel pores were directly calculated from permeation of K$^+$ ions observed in these simulations. The initial 50 ns of the production trajectories were not considered for calculating the K$^+$ ion permeation of the entire 300−400 ns production trajectories (Supplementary Table 2). Each successful count represented an upward ($+Z$) passing K$^+$ ion with the entry point of $Z = -18$ Å and the exit point of $Z = +18$ Å representing the entire pore domain, whereby all snapshots were aligned with the center of mass at the Y1085 origin.

## Reporting summary

Further information on research design is available in the Nature Portfolio Reporting Summary linked to this article.

## Data availability

All data are available from the corresponding authors upon request. The cryo-EM maps have been deposited in the Electron Microscopy Data Bank (EMDB) with the following accession codes: EMD-40496 (TRPM7$_{Closed}$ in MSP2N2 nanodiscs), EMD-40497 (TRPM7$_{Closed}$ in GDN), EMD-40498 (TRPM7-N1098Q$_{open}$), EMD-40499 (TRPM7$_{NTB-open}$), EMD-40500 (TRPM7$_{NTB-closed}$), EMD-40501 (TRPM7$_{VER-closed}$), EMD-40502 (TRPM7-N1098Q$_{VER-closed}$), EMD-40504 (TRPM7-N1098Q$_{NS-closed}$), and EMD-40505 (TRPM7 MHR1-3 domain). The coordinates for the atomic models have been deposited in the Protein Data Bank (PDB) under accession codes 8SI2 [https://doi.org/10.2210/pdb8SI2/pdb] (TRPM7$_{Closed}$ in MSP2N2 nanodiscs), 8SI3 [https://doi.org/10.2210/pdb8SI3/pdb] (TRPM7$_{Closed}$ in GDN), 8SI4 [https://doi.org/10.2210/pdb8SI4/pdb] (TRPM7-N1098Q$_{open}$), 8SI5 [https://doi.org/10.2210/pdb8SI5/pdb] (TRPM7$_{NTB-open}$), 8SI6 [https://doi.org/10.2210/pdb8SI6/pdb] (TRPM7$_{NTB-closed}$), 8SI7 [https://doi.org/10.2210/pdb8SI7/pdb] (TRPM7$_{VER-closed}$), 8SI8 [https://doi.org/10.2210/pdb8SI8/pdb] (TRPM7-N1098Q$_{VER-closed}$), 8SIA [https://doi.org/10.2210/pdb8SIA/pdb] (TRPM7-N1098Q$_{NS-closed}$), and 8SIB [https://doi.org/10.2210/pdb8SIB/pdb] (TRPM7 MHR1-3 domain). The core of the TRPM7$_{Closed}$ model was built using PDB structure 5ZX5 [https://doi.org/10.2210/pdb5ZX5/pdb] as a guide. Source data are provided with this paper.

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

## Acknowledgements

We thank R. Grassucci, Z. Zhang and S. Banerjee (Columbia University Cryo-Electron Microscopy Center) for help with microscope operation and data collection. Some of this work was performed at the Columbia University Cryo-Electron Microscopy Center. We thank Joanna Zaisserer and Anna Erbacher (Walther-Straub Institute, LMU Munich) for their technical assistance. This research was, in part, supported by the National Cancer Institute's National Cryo-EM Facility at the Frederick National Laboratory for Cancer Research under contract 75N91019D00024. A.N. is a Walter Benjamin Fellow funded by the Deutsche Forschungsgemeinschaft (DFG, German Research Foundation)—464295817. V.C. and T.G. were supported by DFG TRR 152 (P15) and GRK 2338 RTG (P10). M.G.K. is supported by the NIH (R01 NS083660, R01 GM128195, R01 AG065594) and NSF (1818213, 1563291) and grants providing time on ANTON2 at the Pittsburgh Supercomputing Center (PSC) (NIH R01GM116961 and D.E. Shaw Research), and XSEDE (NSF ACI-1548562). A.I.S. is supported by the NIH (R01 AR078814, R01 CA206573, R01 NS083660, R01 NS107253).

## Author contributions

K.D.N. made constructs, prepared protein samples, performed planar lipid bilayer recordings, and carried out cryo-EM data processing. K.D.N. and A.N. prepared cryo-EM samples. K.D.N. and A.I.S. analyzed structural data. A.I.S. built molecular models. L.C., T.G. and V.C. performed functional analysis. C.N., D.S.P., and M.G.K. designed computational studies. C.N. performed all MD simulations and MD analysis. D.S.P. carried out water permeation and ion conductivity analysis of MD trajectories. K.D.N., V.C., and A.I.S. wrote the manuscript, which was then edited by all the authors.

## Competing interests

The authors declare no competing interests.
