## [Peer Review File · Nature Communications]

Structural mechanisms of TRPM7 activation and inhibitionReviewers' Comments:

Reviewer #1:

Remarks to the Author:

In this manuscript, Nadezhdin et al. investigated the structural basis of ligand binding and channel activation in mouse TRPM7, a non-selective cation channel that plays a central role in magnesium homeostasis, by combining single-particle cryo-EM, functional analysis, and MD simulation. They used a truncation construct that lacks the C-terminal kinase domain but is still functional, similar to a previously study by Duan et al. However, in comparison to Duan et al, which reported only moderate resolution (3.x Å) TRPM7 structures in the closed state, the cryo-EM structures in the this study were not only of significantly higher quality (2.x angstrom), but were determined in different functional states, including the apo closed state, two open states obtained by a gain-of-function mutation and an agonist, respectively, and two (similar) inhibited states in complex with two different antagonists. Moreover, the agonist binding site found in TRPM7 is novel and has not been seen in other existing structures of the TRPM family.

In summary, this is a fantastic work on an important target TRPM7, providing comprehensive information on ligand recognition, how a single mutation leads to constitutive channel opening, and how agonist binding leads to channel activation. I highly recommend this paper for publication, and only have some minor suggestions and comments.

1. In both open state structures, the pore is restricted by Y1085, which is located in the middle of the pathway. The authors suggest that the side chain hydroxyl group of Y1085 contributes to permeation by forming a hydrogen bonding network with water molecules. While this notion seems to be consistent with the Y1085F leading to a loss-of-function phenotype, can the authors discuss in the manuscript why the Y1985S mutant also leads to a loss-of-function phenotype, assuming that the side chain of serine can also form a hydrogen bonding network with water molecules and is significantly smaller?
2. NTB activation site. The authors should define the linker domain, and show/label it in the corresponding figures.
3. It's interesting that NTB was found in two different sites, with NTB binding to one site leading to channel activation, while binding to the other site had essentially no effect (or perhaps stabilizing the apo closed state). It would be nice if the authors could discuss this observation in more detail. For example, the densities of both NTB molecules should be shown. Do the two NTB molecules have the same conformation? Is it possible that they are enantiomers?
4. The two reported open states show distinct conformational dynamics. Can the authors comment in the manuscript which open state is more likely to occur in a physiological context?
5. The format of the references is inconsistent, e.g., Ref. 20 vs 33.

Wei Lü

Reviewer #2:

Remarks to the Author:

The manuscript 'spontaneous and agonist-induced opening of the divalent cation-selective TRP channel TRPM7' is a very novel and interesting manuscript showing the apo-state structures of TRPM7 of a very high quality resulting in a more complete model of TRPM7, including the regions that were not resolved in previous published structures. In addition, Nadezhdin et al showed convincing structures of the gain-of-function N1098Q variant, showing overlapping VSD, but clear differences in

the pore domain and the positions of the TRP helices. Furthermore, the authors identified for the first time an interaction site for the TRPM7 agonist naltriben and the TRPM7 blockers NS8593 and VER155008. The presented results are very convincing and contain high novelty as the molecular basis of activation and inhibition of TRPM7 and closely related TRPM members was never shown before.

Major comments

- The authors performed mutagenesis to provide evidence that the agonist-induced activation of TRPM7 is mediated by the binding of NTB to the identified region. EC50 values were calculated from concentration-response curves at doses where no plateau was achieved. Therefore data should be interpreted very carefully. Have the authors thought about the effect of the double mutant (LA+NK) to see whether the agonistic effect of NTB on TRPM7 is further reduced? The study would benefit from additional mutagenesis and/or additional experimental tools to investigate the potential ligand-interaction site of NTB.

- In a previous manuscript the authors discussed N1098 as an important residue (N1098Q) that is located in an important regulatory domain of TRPM7. However, this residue is no longer sensitive for the agonist naltriben and is less sensitive for the TRPM7 inhibitor NS8593. Based on these novel insights and the identification of the interaction site, how can the authors explain the lack of sensitivity of the gain-of-function mutation N1098Q towards the channel pharmacology and more specific towards NS8593 as these are placed at different locations?

- Single channel data shown in extended figure 4 are not very convincing (panel 2.5 μM NTB is very variable) as only a representative trace is shown. The authors have to include statistical analysis of multiple traces;

Minor comments

- Do the authors claim that the blocking mechanism by intracellular magnesium is fully independent of the TRPM7 activation mechanism by the agonist naltriben? As a suggestion, it would be of high interest to create the double mutation of LA+NQ or NK+NQ to further prove the evidence that the ligand induced openings mechanism is different from the block by intracellular Mg^{2+} .

- Very recently, a first cryo-EM structure of TRPM3 was published showing the channel in a closed conformation). Please update this manuscript in the introduction.

- The simulation of channel opening between TRPM7 closed – TRPM7-N1098Q is somehow arbitrary, as this movement will not occur under physiological conditions. Do the authors believe that during the shift of high intracellular magnesium towards low intracellular Mg a similar movement will appear? Please discuss this topic further in the manuscript.

- The authors claim to be pioneers in the fact that 'Four different interaction sides, one per monomer of the TRPM7 tetramer, which have not been identified in TRPM channels. However, this statement is incorrect, as previous results of Janssens et al (PMID: 21878524) showed that each channel has four independent and energetically equivalent interaction sites. Please adapt the manuscript.

- Is the interaction site for NTB accessible from both directions (in- and outside the cell?). Would it be possible to further validate the authors' hypothesis via functional inside-out/ outside out patch-clamp experiments?

- The IC50 value of VER for TRPM7 was calculated as 0.23 μM , however for the identification of the antagonist-binding site location, a dose of 100 μM (500 times higher) was added. The authors have to motivate why such a high dose was used and whether the high dose is still channel specific.

- Figure panel C; Please adapt the color code for TRPM7-N1098Q open conform the figure legend.
- Not a fan of a general use of the name 'vanilloid-like site' as none of the TRPM members can be activated by vanilloids. The use of vanilloid-like site could induce confusion for researchers non-familiar to the topic.
- The authors claim that both VER and NS8593 use the same vanilloid-like pocket to inhibit TRPM7 activity. To provide further evidence for this hypothesis, it would be of interest as the authors could provide evidence for this competitive interaction of both antagonists to block the TRPM7 activity.
- Discussion, please insert the references (PMID: 36648066) for the TRPM7-N1098Q, as a similar Asp is identified in TRPM3 causing a spectrum of neurodevelopmental disorders in TRPM3.

Reviewer #3:

Remarks to the Author:

In this manuscript, Nadezhdin and coworkers report beautiful new high-resolution cryo-EM structures of TRPM7 in a closed apo state and in an open conducting state, the latter has not been reported before. The structure of the open form is actually obtained by using either an agonist (naltriben, NTB) or a gain-of-function mutation (N1098Q). In addition to those three structures, the authors also report structures of the TRPM7 (WT and N1098Q) in the presence of two different antagonists, revealing two closed structures and identifying the binding sites of the antagonists. Altogether this paper provides novel information about the permeation mechanism of TRPM7, using a combination of structural biology and molecular simulations, in addition to functional assays.

The authors performed molecular dynamics simulations of the WT apo closed form and of the N1098Q open form (in the absence and presence of a voltage potential). The simulations are long and unequivocally show a difference between both forms; no permeation of water or ion is observed for the apo closed form while the N1098Q mutant (and the form with agonist bound) is permeable to water and ions in the absence of a voltage potential, and shows continuous conductance when the 600 mV potential is applied. The simulations of the N1098Q mutant indicate that is indeed is an open form. All simulations reveal interesting features of the permeation mechanism.

The simulation work is rigorous and solid and altogether brings functional insights to the cryoEM structures. The figures are clear and nicely complement the text. I have one (major) question and a few suggestions that I hope the authors will consider.

(1) Why are monovalent cations used in the simulations? The authors ought to explain to which extent their mechanistic observations with monovalent cations transfer to larger and divalent cations, such as calcium used in the reported experiments.

Further why is K⁺ used when the voltage potential is applied, but Na⁺ in the absence of the voltage potential? The only difference between the two in the force field is presumably the van der Waals parameters.

(2) Few data is provided on the stability of the simulations and the properties of the simulated systems, in particular that of the membrane. The author should provide RMSD plots for the proteins, and report data showing the properties of the lipid bilayers during the long MD simulations (area per lipid, etc.). They should also report on which part of their simulations the data reported in Extended Data Table 2 are calculated. I imagine it is the production runs (0.8-1 microseconds). Yet in the Results section (p.7, l.140), a duration of 2 microseconds is mentioned for the length of the simulation of TRPM-N1098Q. A table summarizing all simulations, the systems compositions and simulations lengths might be useful to include as extended data.

(3) The supplementary videos are insightful and allow the reader to better understand the permeation mechanism. I suggest that the authors provide more information about the time scale of the video, possibly in the video captions, or using a time stamp on the video itself.

(4) I encourage the authors to deposit their trajectories on a repository such as eg. Zenodo.

We are very thankful to Reviewers for their excellent suggestions. We have made changes in the manuscript accordingly with the details outlined in our responses below.

Reviewer #1 (Remarks to the Author):

In this manuscript, Nadezhdin et al. investigated the structural basis of ligand binding and channel activation in mouse TRPM7, a non-selective cation channel that plays a central role in magnesium homeostasis, by combining single-particle cryo-EM, functional analysis, and MD simulation. They used a truncation construct that lacks the C-terminal kinase domain but is still functional, similar to a previously study by Duan et al. However, in comparison to Duan et al, which reported only moderate resolution (3.x Å) TRPM7 structures in the closed state, the cryo-EM structures in the this study were not only of significantly higher quality (2.x angstrom), but were determined in different functional states, including the apo closed state, two open states obtained by a gain-of-function mutation and an agonist, respectively, and two (similar) inhibited states in complex with two different antagonists. Moreover, the agonist binding site found in TRPM7 is novel and has not been seen in other existing structures of the TRPM family.

In summary, this is a fantastic work on an important target TRPM7, providing comprehensive information on ligand recognition, how a single mutation leads to constitutive channel opening, and how agonist binding leads to channel activation. I highly recommend this paper for publication, and only have some minor suggestions and comments.

We thank Dr. Lü for kind assessment of our work.

1. In both open state structures, the pore is restricted by Y1085, which is located in the middle of the pathway. The authors suggest that the side chain hydroxyl group of Y1085 contributes to permeation by forming a hydrogen bonding network with water molecules. While this notion seems to be consistent with the Y1085F leading to a loss-of-function phenotype, can the authors discuss in the manuscript why the Y1985S mutant also leads to a loss-of-function phenotype, assuming that the side chain of serine can also form a hydrogen bonding network with water molecules and is significantly smaller?

This is a good point. Our molecular dynamics simulations demonstrated that in the open state, the Y1085 side chains form a planar, symmetrical arrangement, characterized by a stable $C_{\alpha}-C_{\beta}$ torsion angle (χ_1) of nearly -180° that was maintained throughout runs. With this arrangement, the side chain hydroxyl groups of Y1085 formed a hydrogen bonding network with water molecules in the channel. The lack of hydroxyl groups in Y1085F mutant or their larger separation in Y1085S mutant are likely to disturb this hydrogen bonding network. Therefore, the loss-of-function effects of these mutations may result from the disturbed hydrogen bonding network causing improper hydration/dehydration of permeant cations, thus impeding their conductance through the pore. We have added the corresponding discussion to the text (lines 158-162).

2. NTB activation site. The authors should define the linker domain, and show/label it in the corresponding figures.

We are thankful to Reviewer #1 for noticing this obvious oversight. To be consistent with the previously published data, we changed “linker domain” to “MHR4/Pre-S1 region” and added the corresponding labels to our figures.

3. It's interesting that NTB was found in two different sites, with NTB binding to one site leading to channel activation, while binding to the other site had essentially no effect (or perhaps stabilizing the apo closed state). It would be nice if the authors could discuss this observation in more detail. For example, the densities of both NTB molecules should be shown. Do the two NTB molecules have the same conformation? Is it possible that they are enantiomers?

We have now added a side-by-side comparison of the NTB molecules from the two structures in their density as well as superposition (Supplementary Figure 7, panels g-i). The comparison shows that the NTB molecules from TRPM7_{NTB-open} and TRPM7_{NTB-closed} structures are sterically very similar and do not represent optical isomers (enantiomers). The corresponding discussion has been added to the text (lines 249-251).

4. The two reported open states show distinct conformational dynamics. Can the authors comment in the manuscript which open state is more likely to occur in a physiological context?

We thank the reviewer for this comment. We hypothesize that TRPM7-N1098Q-open represents a spontaneously open state of the channel because it does not require an energetically costly rearrangement of the MHR regions. At the same time, the TRPM7 activity in cells can be fine-tuned by putative endogenous ligands, and in this case, the channel may adopt a conformation similar to the TRPM7-NTB-open structure. Further experiments are needed to address this important question. The corresponding discussion has been added to the text (lines 338-347).

5. The format of the references is inconsistent, e.g., Ref. 20 vs 33.

References formatting has been fixed.

Wei Lü

Reviewer #2 (Remarks to the Author):

The manuscript ‘spontaneous and agonist-induced opening of the divalent cation-selective TRP channel TRPM7’ is a very novel and interesting manuscript showing the apo-state structures of TRPM7 of a very high quality resulting in a more complete model of TRPM7, including the regions that were not resolved in previous published structures. In addition, Nadezhdin et al showed convincing structures of the gain-of-function N1098Q variant, showing overlapping VSD, but clear differences in the pore domain and the positions of the TRP helices. Furthermore, the authors identified for the first time an interaction site for the TRPM7 agonist naltriben and the TRPM7 blockers NS8593 and VER155008. The presented results are very convincing and contain high novelty as the molecular basis of activation and inhibition of TRPM7 and closely related TRPM members was never shown before.

We thank Reviewer #2 for kind words about novelty and quality of our results.

Major comments

- The authors performed mutagenesis to provide evidence that the agonist-induced activation of TRPM7 is mediated by the binding of NTB to the identified region. EC50 values were calculated from concentration-response curves at doses where no plateau was achieved. Therefore data should be interpreted very carefully. Have the authors thought about the effect of the double mutant (LA+NK) to see whether the agonistic effect of NTB on TRPM7 is further reduced? The study would benefit from additional mutagenesis and/or additional experimental tools to investigate the potential ligand-interaction site of NTB.

Unfortunately, we could not examine the effects of NTB in the mM range because of the limited solubility of the compound, and we agree with Reviewer #2 that the impact of point mutations in the NTB-binding site needs to be further clarified. To this end, we analyzed the effects of N752K and L671A mutations using patch-clamp recordings and found that both substitutions reduced TRPM7 sensitivity to 50 and 100 μ M NTB (new Supplementary Fig. 8). As suggested, we also evaluated the double mutant N752K/L671A and observed a further rightward shift of the NTB concentration dependence compared to the concentration dependencies for the corresponding N752K and L671A single mutants (see revised Fig. 3e and Supplementary Table 3). Moreover, we introduced three additional point mutations (S744A, M741A and D670A) of residues in the NTB-binding site (Fig. 3d) and found that each one of them caused a rightward shift of the NTB concentration dependence compared to wild type (revised Fig. 3e and Supplementary Table 4), strongly supporting contribution of these residues to NTB binding.

- In a previous manuscript the authors discussed N1098 as an important residue (N1098Q) that is located in an important regulatory domain of TRPM7. However, this residue is no longer sensitive for the agonist naltriben and is less sensitive for the TRPM7 inhibitor NS8593. Based on these novel insights and the identification of the interaction site, how can the authors explain the lack of sensitivity of the gain-of-function mutation N1098Q towards the channel pharmacology and more specific towards NS8593 as these are placed at different locations?

In our previous publication (Schmidt et al., 2022, PMID: 35389104) we noted that the N1098Q variant did not respond to naltriben. The observation was that the fully activated N1098Q mutant cannot be further activated by the agonist NTB. This finding is supported by our structural data, which shows that under cryo-EM conditions the N1098Q mutant adopts an open conformation. Application of NTB would not open this channel any further. In the case of inhibitors, we found that the N1098Q mutation shifts the equilibrium between the closed and open states of TRPM7 towards more open conformations. As a result, higher concentrations of NS8593 or VER155008 antagonists are required to fully inhibit the channel, which is exactly what we observed (see Supplementary Figure 9).

- Single channel data shown in extended figure 4 are not very convincing (panel 2.5 μ M NTB is very variable) as only a representative trace is shown. The authors have to include statistical analysis of multiple traces;

We performed single-channel recordings mainly to demonstrate that our purified protein is active in planar lipid bilayer membranes, demonstrating a closer link of the structural biology experiment conditions to physiologically relevant behavior of TRPM7. We made sure to avoid overinterpretation of these data in the text. Additional statistical analysis has now been added to Supplementary Fig. 4.

Minor comments

- Do the authors claim that the blocking mechanism by intracellular magnesium is fully independent of the TRPM7 activation mechanism by the agonist naltriben? As a suggestion, it would be of high interest to create the double mutation of LA+NQ or NK+NQ to further prove the evidence that the ligand induced openings mechanism is different from the block by intracellular Mg²⁺.

We agree with Reviewer #2 that regulation of TRPM7 by intracellular Mg²⁺ is extremely important. We noted that our new cryo-EM models agree well with our previous functional experiments (Schmidt et al., 2022, PMID: 35389104), which suggested the crucial role of N1097 in TRPM7 regulation by intracellular Mg²⁺. We have now mentioned this in the text (lines 341-343). Nevertheless, the current study does not present new structural data about the interaction of Mg²⁺ with TRPM7. It would therefore be premature for us to make additional claims regarding the mechanism of block by intracellular Mg²⁺. Along these lines, additional functional analysis of Mg²⁺ effects would also go beyond the scope of the current manuscript.

- Very recently, a first cryo-EM structure of TRPM3 was published showing the channel in a closed conformation). Please update this manuscript in the introduction.

We added the corresponding reference to our manuscript. We also slightly modified Abstract and Introduction sections to better fit the existing literature on TRPM channels.

- The simulation of channel opening between TRPM7 closed – TRPM7-N1098Q is somehow arbitrary, as this movement will not occur under physiological conditions. Do the authors believe that during the shift of high intracellular magnesium towards low intracellular Mg a similar movement will appear? Please discuss this topic further in the manuscript.

We noted that the obtained cryo-EM data for WT and N1098Q TRPM7 are well consistent with the proposed model for Mg²⁺-mediated inhibition of TRPM7 spontaneous activity (Schmidt et al., 2022, PMID: 35389104) a In the current manuscript, we do not provide any novel structural results related to TRPM7 regulation by intracellular Mg²⁺. Correspondingly, we refrain from making additional comments in the text on the topic that is outside the scope of this manuscript.

- The authors claim to be pioneers in the fact that 'Four different interaction sides, one per monomer of the TRPM7 tetramer, which have not been identified in TRPM channels. However, this statement is incorrect, as previous results of Janssens et al (PMID: 21878524) showed that each channel has four independent and energetically equivalent interaction sites. Please adapt the manuscript.

We are grateful to Reviewer #2 for this comment. We have modified the sentence mentioned above to better clarify our point that the corresponding binding site had not been identified in TRPM channels before (Lines 210-212).

- Is the interaction side for NTB accessible from both directions (in- and outside the cell?). Would it be possible to further validate the authors' hypothesis via functional inside-out/ out side out patch-clamp experiments?

NTB is a small hydrophobic molecule that is expected to rapidly cross biological membranes, similar to other drug-like molecules. Consistent with this view, the external application of NTB allows TRPM7 activation in both Ca^{2+} imaging and patch-clamp experiments. Hence, we are unsure that applying naltriben in inside-out vs outside-out configurations will produce well-distinguished effects. In addition, we performed extensive functional analysis of TRPM7 combined with site-directed mutagenesis, which are highly consistent with our cryo-EM data and provide strong support to the location of the identified NTB binding site.

- The IC_{50} value of VER for TRPM7 was calculated as 0.23 μM , however for the identification of the antagonist-binding site location, a dose of 100 μM (500 times higher) was added. The authors have to motivate why such a high dose was used and whether the high dose is still channel specific.

Usage of high concentration is necessary to ensure high occupancy of binding sites in structural studies. To satisfy this requirement, the concentration of ligand in the sample should be not only higher than the IC_{50} value but also higher than the concentration of the protein, which is $\sim 30 \mu\text{M}$ in this case.

- Figure panel C; Please adapt the color code for TRPM7-N1098Q open conform the figure legend.

We are grateful to Reviewer #2 for this comment. We made a corresponding correction in the figure legend.

- Not a fan of a general use of the name 'vanilloid-like site' as none of the TRPM members can be activated by vanilloids. The use of vanilloid-like site could induce confusion for researchers non-familiar to the topic.

We can of course call this site with the name of TRPM7-specific ligand but structurally it will remain at the position homologous to the vanilloid site in TRPV channels. Given the similarity of transmembrane domain in all TRP channels, it is useful to have the universal ligand binding site nomenclature, so that all TRP channels can be easily compared in this respect. This particular name is used based on the historical reasons, because this site was one of the first to be discovered and described in detail. We are also following the nomenclature that we developed in our recent review on ligand binding sites ([doi:10.3389/fphar.2022.900623](https://doi.org/10.3389/fphar.2022.900623)).

- The authors claim that both VER and NS8593 use the same vanilloid-like pocket to inhibit TRPM7 activity. To provide further evidence for this hypothesis, it would be of interest as the authors could provide evidence for this competitive interaction of both antagonists to block the TRPM7 activity.

The 'golden standard' approach for addressing such question is a competitive ligand-binding assay with radioactively labelled drugs, which are not available yet. Therefore, we further studied the TRPM7-M991A mutant, which showed no sensitivity to 0.1-10 μM VER155008 (Fig. 5e) but showed the usual response to NS8593 (Fig. 5g). We assumed that VER155008 retains its ability to interact with TRPM7-M991A, although without functional effect on the channel. If this assumption is correct and both compounds act by binding to the same site on TRPM7, the presence of 10 μM VER155008 should cause a rightward shift in the NS8593 concentration dependence. The results of this experiment are shown in the new panel j of Supplementary Fig. 7. Consistent with binding of NS8593 and VER155008 to the same site, the NS8593 concentration dependence for TRPM7-M991A is rightward shifted in the presence of 10 μM VER155008.

- Discussion, please insert the references (PMID: 36648066) for the TRPM7-N1098Q, as a similar Asp is identified in TRPM3 causing a spectrum of neurodevelopmental disorders in TRPM3.

We have now added the suggested reference (PMID: 36648066).

Reviewer #3 (Remarks to the Author):

In this manuscript, Nadezhdin and coworkers report beautiful new high-resolution cryo-EM structures of TRPM7 in a closed apo state and in an open conducting state, the latter has not been reported before. The structure of the open form is actually obtained by using either an agonist (naltriben, NTB) or a gain-of-function mutation (N1098Q). In addition to those three structures, the authors also report structures of the TRPM7 (WT and N1098Q) in the presence of two different antagonists, revealing two closed structures and identifying the binding sites of the antagonists. Altogether this paper provides novel information about the permeation mechanism of TRPM7, using a combination of structural biology and molecular simulations, in addition to functional assays.

The authors performed molecular dynamics simulations of the WT apo closed form and of the N1098Q open form (in the absence and presence of a voltage potential). The simulations are long and unequivocally show a difference between both forms; no permeation of water or ion is observed for the apo closed form while the N1098Q mutant (and the form with agonist bound) is permeable to water and ions in the absence of a voltage potential, and shows continuous conductance when the 600 mV potential is applied. The simulations of the N1098Q mutant indicate that is indeed is an open form. All simulations reveal interesting features of the permeation mechanism.

The simulation work is rigorous and solid and altogether brings functional insights to the cryoEM structures. The figures are clear and nicely complement the text. I have one (major) question and a few suggestions that I hope the authors will consider.

We thank Reviewer #3 for kind words about our work.

(1) Why are monovalent cations used in the simulations? The authors ought to explain to which extent their mechanistic observations with monovalent cations transfer to larger and divalent cations, such as calcium used in the reported experiments.

The choice to use monovalent ions for the simulations stems from technical reasons and limitations of the molecular mechanics method. With modern forcefields, the permeation of monovalent cations in the presence of an electric field is modeled more reliably than permeation of divalent cations. Since TRPM7 permeates monovalent cations in the absence or low concentration of divalent cations, the goal of the simulations was to demonstrate that the resolved structures correspond to an open channel. Simulating monovalent ion permeation is suitable for answering this question.

Further why is K⁺ used when the voltage potential is applied, but Na⁺ in the absence of the voltage potential? The only difference between the two in the force field is presumably the van der Waals parameters.

Indeed, the only difference between the modeled Na⁺ and K⁺ cations is the Lennard-Jones parameters. However, the slightly larger sigma (radius) of the K⁺ ion compared to the Na⁺ ion results in a very different behavior of the two ions with respect to coordinating ligands in the first solvation shell.

Because of its size and slightly weaker ligand attraction, K^+ can accommodate 6-8 ligands in the first solvation shell and exchange ligands strictly by an associative mechanism. This makes it a more forgiving ion while permeating a narrow and fairly hydrophobic environment of the ion channel. In simulations, this will result in better channel permittivity of K^+ for the purpose of acquiring statistically significant values of the permeation. Such behavior has been shown in prior work by us and others in different ion channels (Yelshanskaya et al., Nature 2022, PMID 35444281; Biedermann et al., PNAS 2021, PMID 33602810).

Due to the difficulty of directly simulating ion permeation in the presence of the applied voltage and insufficiency of water and ion parameterization in standard molecular mechanics models, the ion permeation results can only be considered semi-quantitative. Therefore, the slight difference in permeation rates of Na^+ and K^+ under various experimental conditions will not be quantitatively assessed. While we expect that the order of permeation rate for various cations can be predicted correctly, demonstrating the permeation of any of the ions is sufficient to demonstrate how the channel forms an open pore, regardless of the specific ion.

(2) Few data is provided on the stability of the simulations and the properties of the simulated systems, in particular that of the membrane. The author should provide RMSD plots for the proteins, and report data showing the properties of the lipid bilayers during the long MD simulations (area per lipid, etc..).

As the reviewer suggested, we have included a new Supplementary Fig.5 with protein RMSD plots for the representative systems of TRPM7closed, TRPM7-N1098Q, and TRPM7-NTB-open. We also include a new Supplementary Table 2 that includes the average RMSD values for all simulated systems.

The below plot shows that area per lipid (calculated for the top leaflet of TRPM7-N1098Q, TRPM7-closed, and TRPM7-NTB-open systems using GridMAT-MD) is stable during MD simulations. However, it is difficult to reliably calculate the lipid properties for our system as this system has a high protein to lipid ratio and the protein is irregularly shaped.

The simulated protein in the membrane viewed a) parallel to the membrane b) from the top c) from the bottom.

Area per lipid calculated for the top leaflet of TRPM7-N1098Q, TRPM7-closed, and TRPM7-NTB-open systems.

They should also report on which part of their simulations the data reported in Extended Data Table 2 are calculated. I imagine it is the production runs (0.8-1 microseconds). Yet in the Results section (p.7, l.140), a duration of 2 microseconds is mentioned for the length of the simulation of TRPM-N1098Q. A table summarizing all simulations, the systems compositions and simulations lengths might be useful to include as extended data.

Per Reviewer #3 suggestion, we have included Supplementary Table 2 summarizing all simulations. We used the production runs of all systems to calculate the data reported in Supplementary Table 3 (previously named Extended Data Table 2). The 2 μ s of simulations described in the text refers to the total simulation length of two replicas of the TRPM7-N1098Q system ($\sim 1 \mu$ s each). We have revised the text (and the caption for Supplementary Table 3).

We added the following statement to the the caption for Supplementary Table 3:

“Water permeation and ion conductance were computed using production trajectories of all systems as described in Methods. For systems with two replicas, the reported values are averaged over two simulations.”

(3) The supplementary videos are insightful and allow the reader to better understand the permeation mechanism. I suggest that the authors provide more information about the time scale of the video, possibly in the video captions, or using a time stamp on the video itself.

We have added the corresponding information to the video captions:

Supplementary Video 1: The length of the trajectory shown is 4.8 ns.

Supplementary Video 2: The length of the trajectory shown is 45 ns.

(4) I encourage the authors to deposit their trajectories on a repository such as eg. Zenodo.

All MD trajectories are available from the authors upon request. We will also look into depositing them to Zenodo.

Reviewers' Comments:

Reviewer #1:

Remarks to the Author:

The authors have addressed my comments well, and I recommend publication of this paper.

Reviewer #2:

Remarks to the Author:

The authors were able to convincingly address most of my concerns.

Reviewer #3:

Remarks to the Author:

The authors have answered all my questions and comments, and clarified my concerns. I recommend the publication of their work without any reservation.